# Donor transcription suppresses D-loops in *cis* and promotes genome stability

Yasmina Djeghmoum & Aurèle Piazza 📧

## Abstract

**DNA is a substrate for competing protein-mediated activities. Whether and how transcription and the synaptic steps of recombination collide or are coordinated has not been investigated. Here, using a controlled break induction system and physical detection of D-loop DNA joint molecules in *S. cerevisiae*, we show that donor transcription by RNA polymerase II strongly and acutely suppresses D-loops *in cis*. The extent of this suppression depends on the orientation of transcription, suggesting the preferential usage of one end for the repair of DNA break in transcribed regions. Transcription-mediated D-loop suppression does not rely on endogenous transcription factors, the RNA product, or RNA:DNA hybrids. It is independent of, and can be more potent than the conserved *trans* D-loop-disruption factors Sgs1-Top3-Rmi1[BLM-TOPO3α-RMI1/2], Mph1[FANCM], and Srs2. This transcription-mediated control promotes genome maintenance by inhibiting ectopic recombination and multi-invasion-induced rearrangements, while authorizing allelic inter-homolog repair. These findings reveal the prioritization between two universal DNA-dependent processes and its role in promoting genome stability.**

**Keywords** Recombination; Transcription; D-Loop; Genome Stability; Paralogous Genes
**Subject Categories** Chromatin, Transcription & Genomics; DNA Replication, Recombination & Repair

## Introduction

Various protein-dependent processes, such as transcription, replication, and recombination, compete for the same DNA substrate and must be coordinated. Transcription has long been known to stimulate spontaneous homologous recombination (HR) in prokaryotes and eukaryotes by interfering with DNA replication, despite elaborate mechanisms that coordinate their deployment and mitigate conflicts (recently reviewed in (Goehring et al, 2023; Browning and Merrikh, 2024)). Comparatively much less is known about how transcription and HR are coordinated, and how putative priority rules impact genome stability.

HR is a high-fidelity DNA double-strand break (DSB) repair pathway that uses an intact homologous dsDNA molecule as template. It entails the formation of a metastable DNA joint molecule called a Displacement loop (D-loop), which consists of a heteroduplex DNA (hDNA) region, a displaced strand and two non-identical 5′ and 3′ strand exchange junctions (Wright et al, 2018). Their extension by a DNA polymerase commits to repair, while their disruption reinitializes the search for a homologous donor and eliminates toxic joint molecules. This reversibility, enforced by several conserved ancillary HR factors (Sgs1-Top3-Rmi1[BLM-TOPO3a-RMI1/2], Mph1[FANCM], and Srs2), imparts robustness in donor selection and contributes to the high fidelity of DSB repair by HR, which can otherwise lead to repeat-mediated chromosomal rearrangements (Savocco and Piazza, 2021).

Intuitively, transcription at the donor site appears incompatible with the co-occurrence of a D-loop, but priority rules between the two have largely remained unexplored. This is mainly due to (i) technical limitations in detecting D-loops in cells, and (ii) difficulties in disentangling the role of transcription in generating recombinogenic lesions from that of regulating their repair. Evidence in budding yeast and human cells hints at an inhibitory role of transcription at synaptic or post-synaptic HR steps. First, transcriptional activity biases the repair outcome of meiotic inter-homolog recombination towards non-crossovers in humans (McVicker and Green, 2010; Pouyet et al, 2017; Palsson et al, 2025). In budding yeast, early work quantifying spontaneous recombination rates between hetero-alleles with varying transcriptional levels also suggested that, besides generating recombinogenic lesions, transcription could also inhibit their repair (Saxe et al, 2000). More recently, initiation of break-induced replication (BIR) was shown to be impaired by highly-transcribed RNA Polymerase II (hereafter RNA Pol II) genes present in head-on orientation in budding yeast (Liu et al, 2021; Uribe-Calvillo et al, 2022). BIR is a non-canonical, conservative and unstable Rad51-dependent replication process that takes place in the context of a D-loop structure (Davis and Symington, 2004; Lydeard et al, 2007; Smith et al, 2007; Wilson et al, 2013; Mayle et al, 2015; Donnianni and Symington, 2013; Saini et al, 2013; Donnianni et al, 2019; Liu et al, 2021). It suggested a dominance of transcription over recombination, at least in the initial elongation step of this low-fidelity HR sub-pathway. However, how transcription might impinge on the core synaptic steps of canonical HR (i.e., on donor invasion and D-loop metabolism) remains unknown.

Université de Lyon, ENS de Lyon, Université Claude Bernard, CNRS UMR5239, Laboratoire de Biologie et Modélisation de la Cellule, 46 Allée d'Italie, 69007 Lyon, France.
📧 E-mail: aurele.piazza@ens-lyon.fr

Here, using a site-specific DSB induction system and molecular assays for D-loop detection in *S. cerevisiae*, we show that donor transcription by RNA Pol II strongly suppresses D-loops in *cis* in an orientation-dependent manner. We delineate its requirement and relationship with previously characterized D-loop disruption activities, and demonstrate its involvement in suppressing ectopic recombination and repeat-mediated genome rearrangements. These findings reveal the prioritization between two universal DNA-dependent processes and its function in genome maintenance.

## Results

### Experimental system

We used a well-established experimental system in haploid *S. cerevisiae* cells in which a site-specific DSB can be rapidly induced upon overexpression of the *HO* gene (Piazza et al, 2018, 2019). The HO cut-site (HOcs) at the *URA3* locus on chr. V is flanked on the left by a region of homology to a "donor" site present at the *LYS2* locus on chr. II (Fig. 1A). The right side has no donor, which purposefully precludes repair steps downstream of D-loop extension. The levels of D-loop joint molecules formed by the left DSB end at, and extended from this donor, can be quantified over a ~8 h period using the proximity ligation-based D-loop-capture (DLC) and D-loop extension (DLE) assays, respectively (Piazza et al, 2018, 2019; Reitz et al, 2022). DLC uniquely requires in vivo interstrand DNA crosslinking with psoralen, unlike DLE. We used an improved DLC protocol in which psoralen was de-crosslinked prior to qPCR, enabling absolute determination of the amount of D-loops in a cell population (Reitz et al, 2022). In this system, >90% of DNA molecules are cut within 1 h of *HO* expression induction, D-loops are first detected at 2 h, peak at 3-4 h, and are extended from 4 to 8 h post-DSB induction (Piazza et al, 2019). Scoring D-loops at 2 h, thus focuses on D-loops prior to their extension.

### Transcription of the donor reduces D-loop levels

In order to investigate the effect of the RNA Pol II-dependent transcriptional activity at the donor site on D-loop metabolism, we placed the 1-kb-long "L" donor under the control of promoters that drive either no or minimal transcription (*pDMC1* and *pLYS2*) or high transcription (*pTDH3* and *pGAL1*) in our culture conditions (Fig. 1A). The promoter is not part of the homology region, and is oriented so that the same DNA strand is a template for transcription and recombination, which we refer to as the "co-directional" orientation (Fig. 1B). The expected transcriptional activity conferred by these promoters was verified by ChIP-qPCR of RNA Pol II and RT-qPCR with these different constructs (Fig. EV1A,B). Donor transcription did not affect the expression of the downstream gene *RAD16* either (Fig. EV1B), and had no indirect effect on DSB formation on chr. V (Fig. EV1C).

Strikingly, D-loop levels detected 2 h post-DSB induction were approximately 7-fold lower when the donor was transcribed than when it was not (Fig. 1C). The D-loop level was inversely correlated with the donor transcriptional level (Fig. EV1D). This inhibition was observed at all time points up to 8 h post-DSB induction

(Fig. 1D,E), indicating that the inhibition observed at 2 h was not due to a delay in D-loop formation, but reflected a reduced steady-state D-loop level. Similar results were obtained with an independent construct bearing 2 kb of homology (Figs. 1D,E and EV1E). Consistently, the downstream step of D-loop extension, scored 6 h post-DSB induction, was inhibited approximately sevenfold upon donor transcription (Fig. 1F). These observations indicate that transcription of the donor interferes with the synaptic steps of recombination, which prevents the downstream step of D-loop extension. Transcription may prevent D-loop formation and/or cause their disruption.

### Transcription suppresses D-loops acutely

In order to gain insights into the mechanism of transcription-dependent D-loop suppression, we sought to determine how quickly D-loops responded to transcriptional changes at the donor. To this end, we placed the donor under the control of a copper-inducible *pCUP1* promoter (Labbé and Thiele, 1999). D-loops were left to form in the absence of copper for ~110 min post-DSB induction, and transcription was triggered upon copper addition ~5–10 min prior to DNA crosslinking and D-loop detection (Figs. 2A and EV2A). This short transcriptional induction was sufficient to cause a dose-dependent ~2.5 to 11-fold drop in D-loop levels (Fig. 2A). D-loops formed downstream of the silent *pDMC1* promoter were not affected by copper addition, ruling out an indirect effect of copper on D-loop metabolism (Fig. 2A).

Conversely, we determined the kinetics of D-loop recovery following the glucose-induced shut-off of transcription at a *pGAL1* promoter (Nehlin et al, 1991) (Fig. 2B). D-loops were recovered within 15 min post-glucose addition and remained equivalent to the non-transcribed *pDMC1* donor afterwards (Figs. 2B and EV2B).

These rapid responses to transcriptional activation and shutoff suggest that transcription directly undermines the stability of D-loops (or of the precursor synaptic complex) in *cis*. In the following sections, we examine this possibility by investigating the role of secondary consequences of transcriptional activity on D-loop levels.

### Transcription suppresses D-loops in *cis*

To directly address whether transcription of the donor exerts its effect in *cis*, or whether it acts in *trans* through its RNA product, we determined D-loop formation at the level of two competing donors, only one of which is transcribed. If the transcription exerts its inhibitory effect in *trans*, D-loop levels should be reduced at both donors. If transcription acts in *cis*, D-loop levels should only be reduced at the transcribed donor, and may even redirect D-loop formation onto the other, non-transcribed donor it competes with.

We used a previously characterized system with one donor located on the same chromosome as the DSB site (intra donor) and one located on chr. II (inter donor) (Piazza et al, 2021) (Fig. EV2C). We confirmed with our improved DLC protocol (Reitz et al, 2022) that D-loops formed with a ~20-fold preference at the intra over the inter donor in the absence of transcription (Piazza et al, 2021) (Fig. EV2C,D). Presence or absence of an inter donor did not detectably affect intra D-loop formation (Fig. EV2C). On the contrary, presence of the intra donor caused a ~4- to ~20-fold

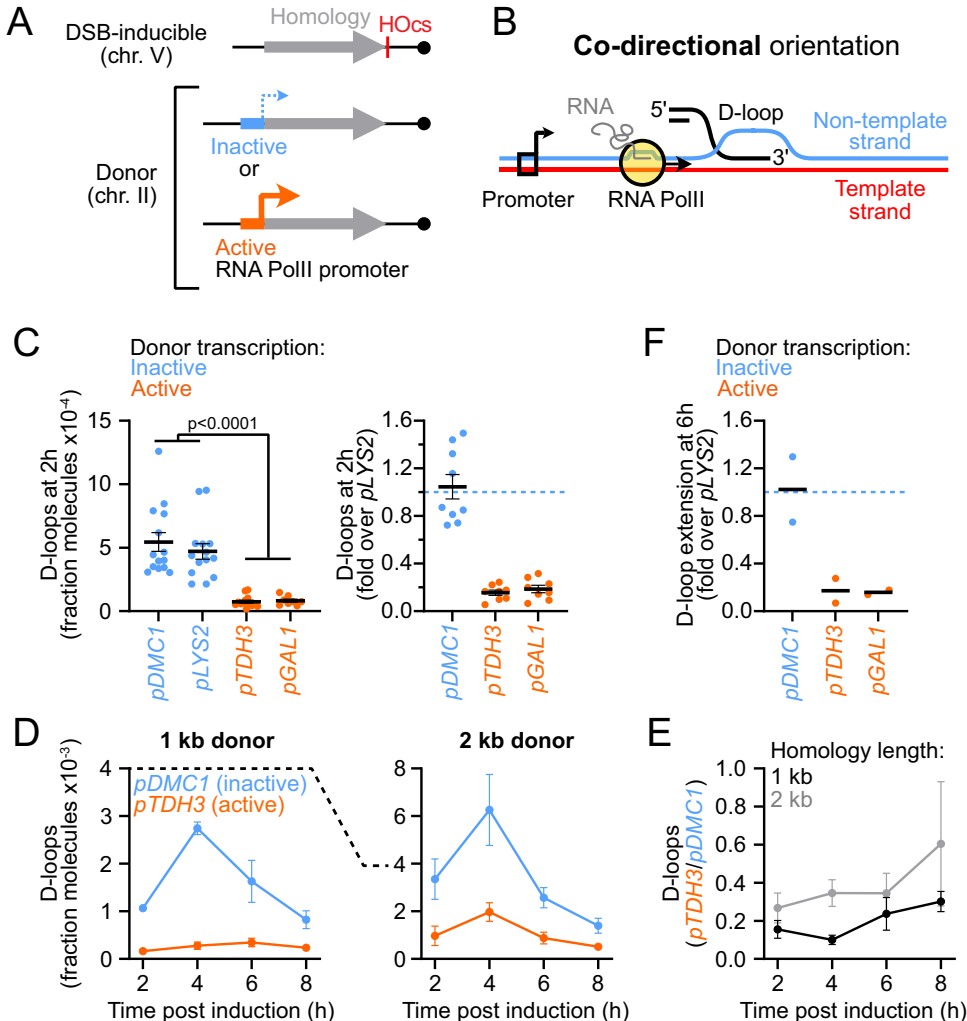

**Figure 1. Co-directional donor transcription suppresses D-loops.**

(A) Experimental system with the donor transcribed in the co-directional orientation. (B) Depiction of RNA Pol II relative to the D-loop in the co-directional orientation. (C) Left: D-loops detected 2 h post-DSB induction with transcriptionally inactive (*pDMC1* and *pLYS2*) and active (*pTDH3* and *pGAL1*) donors (strains APY502, APY867, APY725, and APY724, respectively). Right: D-loop levels relative to that of a strain bearing a transcriptionally inactive *pLYS2* donor scored in parallel. *P* values were computed using a Mann–Whitney Wilcoxon test. (D) D-loops detected following DSB induction with two homology length at transcriptionally inactive and active donors (strains APY502, APY867, APY354, and APY1180). (E) D-loop inhibition upon donor transcription, determined from data in (D). (F) D-loop extension at 6 h post-DSB induction, normalized to that of a strain bearing a transcriptionally inactive *pLYS2* donor scored in parallel. Strains are the same as in (C). (C–F) Data points show individual biological replicates (*n*). Bars show mean ± SEM. Source data are available online for this figure.

reduction in inter D-loops (Fig. EV2E). Hence, the presence of the intra donor outcompetes the inter donor for D-loop formation.

Transcription of the intra donor caused a significant 1.8-fold decrease of intra D-loops, and led to a 2.6-fold increase of inter D-loops (Fig. 2C). It resulted in a 3.7-fold decrease in the intra/inter donor preference (Fig. 2D). Conversely, transcription of the inter donor caused a 2.4-fold decrease of inter D-loops without detectably affecting intra D-loops, which led to a significant 2.8-fold increase in the intra/inter donor preference (Fig. 2C,D). These results establish that transcription specifically reduces D-loop levels at the transcribed donor, demonstrating that transcription suppresses D-loops in *cis*. Furthermore, this *cis*-acting inhibition can redirect D-loop formation onto the non-transcribed donor, swinging the intra/inter donor preference up to tenfold (Fig. 2D).

## Transcription suppresses D-loops independently of RNA:DNA hybrids

RNA:DNA hybrids, containing three-stranded structures called R-loops, may form co-transcriptionally in *cis* at certain highly-transcribed genomic loci (Gómez-González and Aguilera, 2021). Low-level RNA:DNA hybrids could be detected at the locus used as a donor in our system (i.e., *LYS2*) by H-CRAC in mutant contexts (Aiello et al, 2022) or by DRIP-qPCR when it was artificially overexpressed (Mérida-Cerro et al, 2024).

In order to address whether RNA:DNA hybrids were involved in transcription-dependent D-loop inhibition, we evaluated D-loop levels in various contexts reported either to (i) to eliminate RNA:DNA hybrids upon overexpression of RNAseH1 (Wahba et al, 2011) or

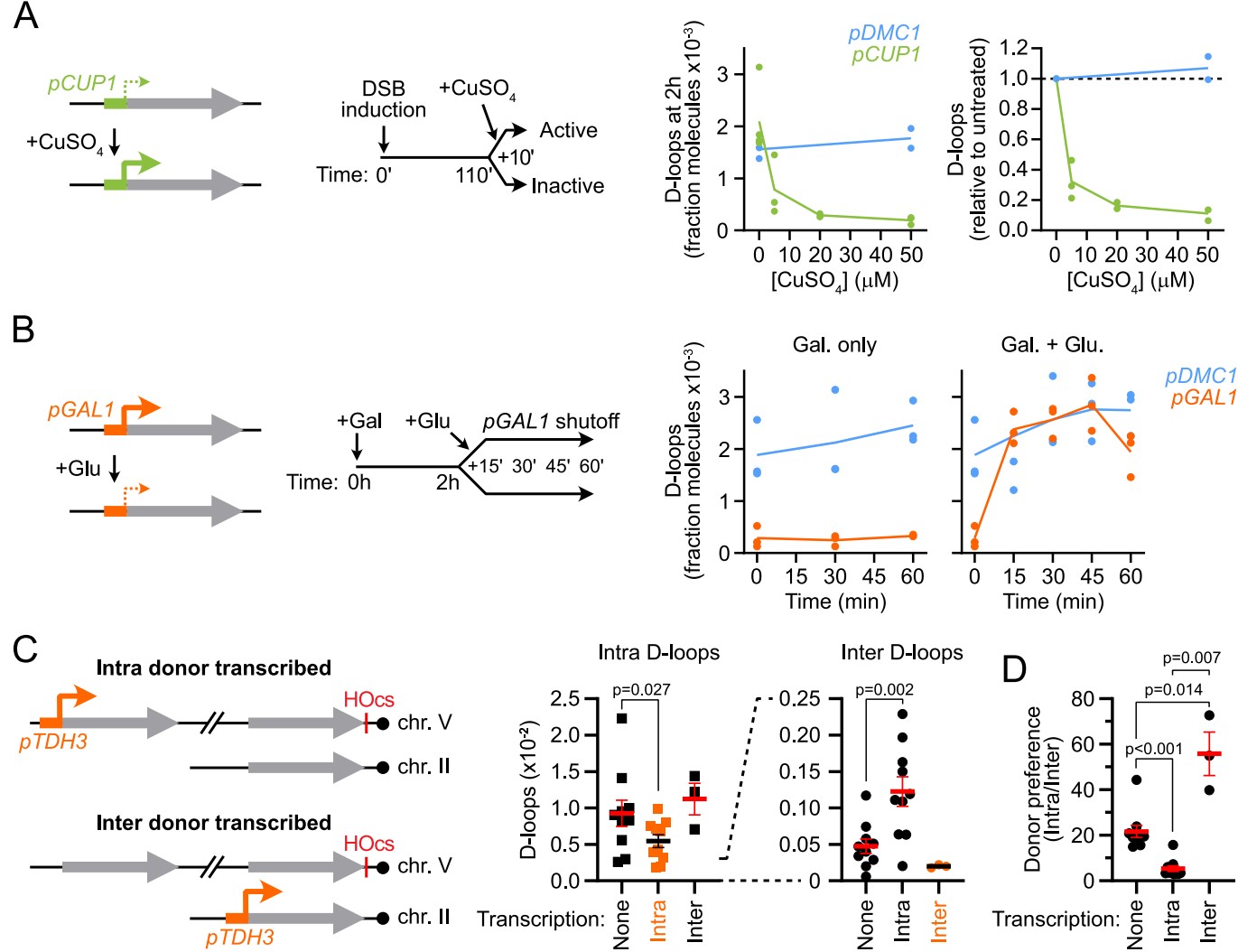

**Figure 2. Donor transcription suppresses D-loops in *cis*.**

(A) Copper-induced transcriptional activation at the donor from the *pCUP1* promoter ~10 min prior to D-loop detection causes a dose-dependent D-loop loss (APY503). The control *pDMC1* promoter is not copper-responsive (APY502). (B) Kinetics of D-loop restoration following transcriptional shutdown at the donor from the *pGAL1* promoter upon glucose addition (APY725). The control *pDMC1* promoter is not sensitive to the carbon source (APY502). (C) Donor transcription inhibits D-loop formation in *cis*. D-loops formed at the intra donor (left) and the inter donor (right) in the absence of donor transcription (APY809), upon transcription of the intra donor (APY1587), or upon transcription of the inter donor (APY1709). *P* values were computed using a Mann–Whitney Wilcoxon test (None vs. Intra-paired; None vs. Inter-unpaired). (D) Fold preference for D-loop formation at the intra over the inter donor. From data in (C). *P* values were computed using a Mann–Whitney Wilcoxon test. (A–D) Data points show individual biological replicates (*n*). Bars show mean ± SEM. Source data are available online for this figure.

(ii) to exacerbate their formation and/or stability, in mutants of the THO complex (*mft1Δ*) or of the transcription elongation factor TFIIS (Huertas and Aguilera, 2003; San Martin-Alonso et al, 2021). RNAseH1 was overexpressed from a *pGAL1* promoter on a multi-copy 2μ plasmid together with DSB induction upon galactose addition. RNAseH1 overexpression did not significantly affect D-loop levels, neither with the *pDMC1* nor the *pTDH3* constructs (Figs. 3A and EV3A). Likewise, the THO complex *mft1Δ* mutant and the *tfiisΔ* mutant did not cause a reduction in D-loop levels (Fig. 3B). These observations indicate that RNA:DNA hybrids are not involved in suppressing D-loops at highly-transcribed genes.

The *tfiisΔ* mutant actually exhibited a modestly reduced transcription-dependent D-loop inhibition, at the limit of statistical

significance (*p* = 0.055, two-tailed unpaired Mann–Whitney test, Fig. 3B). Given the role of TFIIS in promoting transcription elongation by RNA Pol II (Sigurdsson et al, 2002; Zatreanu et al, 2019), this observation suggests that D-loop suppression requires efficient transcription across the donor.

## Transcription suppresses D-loops independently of peripheral nuclear delocalization and endogenous transcription initiation factors

Transcriptional activation can lead to the delocalization of the locus from the nuclear lumen to the nuclear periphery (Casolari et al, 2004; Brickner and Walter, 2004). However such delocalization has

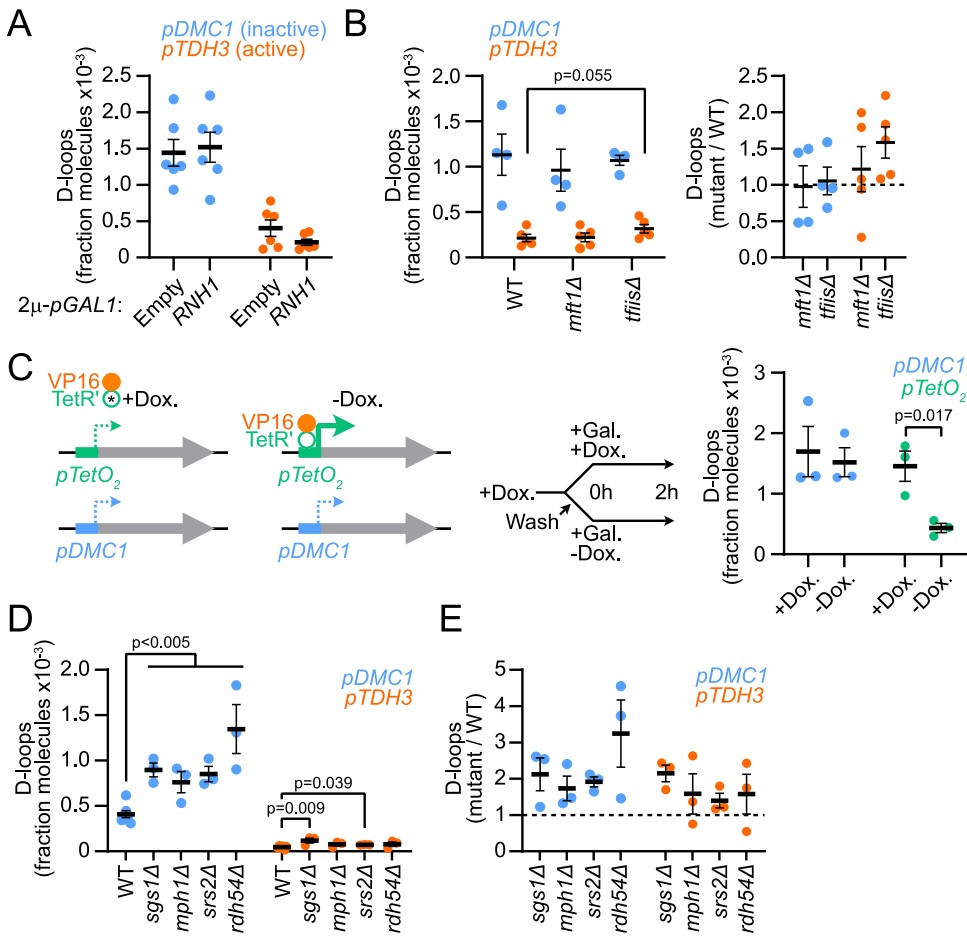

**Figure 3. Genetic determinants of transcription-mediated D-loop suppression.**

(A) D-loop levels are unaffected by RNAseH1 overexpression, irrespective of the transcriptional status of the donor (APY1272, APY1274, APY1278, and APY1280). No significant differences were observed upon *RNH1* overexpression. (B) Left: D-loop levels at transcriptionally inactive and active donors in WT (APY502 and APY725), *mft1Δ* (APY1114 and APY1116) and *tfiisΔ* (APY1485 and APY1487) strains. Right: relative mutant values compared to a WT strain assayed in parallel. *P* values were computed using a Mann–Whitney Wilcoxon test. (C) Left: Rationale and experimental scheme for transcriptional activation by a heterologous TetR-VP16 construct. Right: D-loop levels (APY1292 and APY1294). *P* values were computed using a Student *t*-test. (D) D-loop levels at transcriptionally inactive and active donors in WT (APY502 and APY725), *sgs1Δ* (APY824 and APY795), *mph1Δ* (APY789 and APY796), *srs2Δ* (APY791 and APY798), and *rdh54Δ* (APY793 and APY799) strains. *P* values were computed using a Student *t*-test. (E) Mutant values compared to a WT strain assayed in parallel, from data in (D). (A–E) Data points show individual biological replicates (*n*). Bars show mean ± SEM. Source data are available online for this figure.

been reported to occur ~15 min post-transcriptional induction at the earliest (Randise-Hinchliff et al, [2016]), while less than 10 min of transcriptional induction was sufficient to cause full D-loop disruption (Fig. 2A). Furthermore, *pGAL1* retains its peripheral nuclear localization for >14 h post-transcriptional shutoff with glucose (Sood et al, [2017]), yet D-loops we recovered in less than 15 min in these conditions (Fig. 2B). These results indicate that the peripheral nuclear localization of highly-transcribed genes is not implicated in D-loop suppression.

In order to confirm this independence, and address the role of endogenous transcription initiation factors in mediating D-loop suppression, we placed the donor under the control of a heterologous doxycycline-responsive dual activator/repressor system (Bellí et al, [1998]). In the absence of doxycycline, the TetR' DNA binding domain of *E. coli* fused to the transcription activator VP16 from the SV40 virus drives transcription from a *pTetO2* promoter, placed upstream of the donor (Fig. 3C), while the TetR-

Ssn6 fusion suppresses transcription otherwise. This VP16 fusion strongly stimulates transcription in budding yeast while retaining the transcribed locus in the nuclear lumen (Sadowski et al, [1988]; Garí et al, [1997]; Taddei et al, [2006]). Cells grown in the presence of doxycycline (i.e., in which TetR'-VP16 does not associate to its *TetO* target) exhibited similar D-loop levels whether the donor was under the control of the *pTetO2* or the control *pDMC1* promoter (Fig. 3C). Transcriptional activation from the *pTetO2* promoter upon doxycycline removal caused a 3.5-fold D-loop loss (Fig. 3C). D-loop levels remained unaffected with the *pDMC1* promoter, ruling out indirect effects of doxycycline or of the TetR-VP16 construct on early recombination steps. These results confirm that transcription-mediated D-loop inhibition occurs independently of the delocalization of the transcribed locus at the nuclear periphery. It also shows that such inhibition can be triggered by a heterologous transcription initiation factor, and is thus independent of any specific endogenous transcription initiation factor.

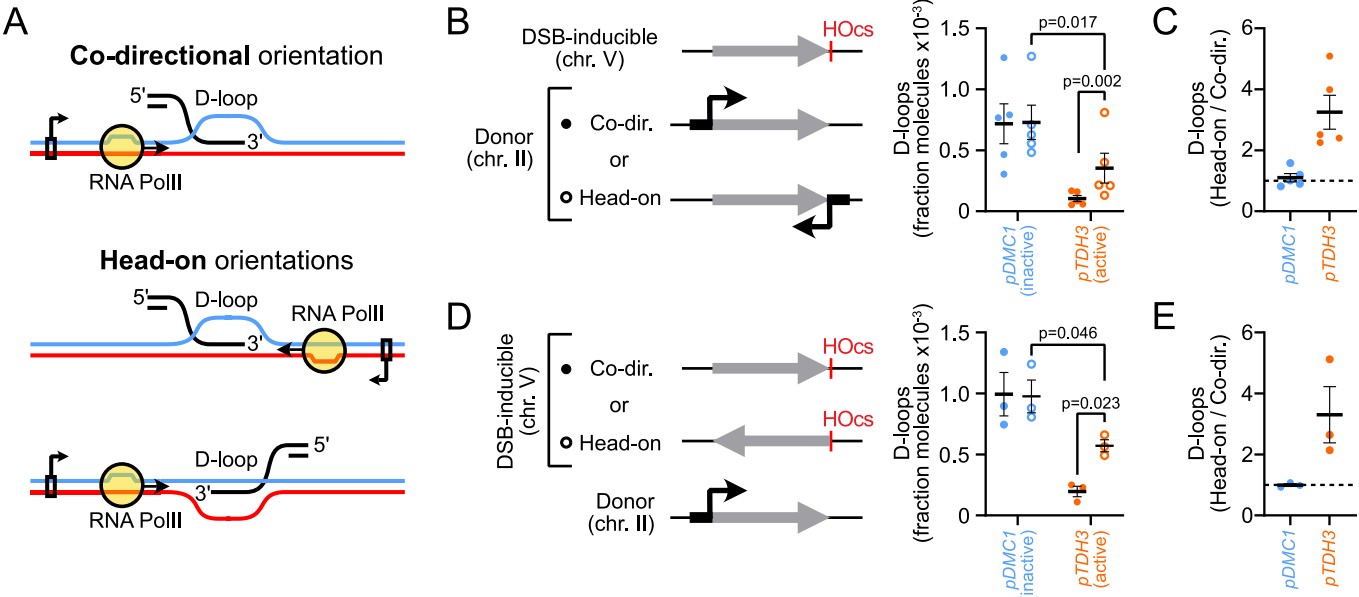

**Figure 4. Effect of transcription directionality on D-loop suppression.**

(A) Depiction of RNA Pol II relative to the D-loop in the co-directional and head-on orientations. (B) Inverting the transcription directionality at the donor by repositioning the promoter partly rescues D-loop levels when transcription is active (APY502, APY941, APY725, and APY999). P values were computed using a paired Student t-test. (C) Fold difference in D-loop levels in the head-on compared to the co-directional orientation. Data from (B). (D) Inverting the transcription directionality at the donor by inverting the homology sequence near the DSB partly rescues D-loop levels when transcription is active (APY502, APY1731, APY725, and APY1733). P values were computed using a paired Student t-test. (E) Same as (C), with data from (D). (B–E) Data points show individual biological replicates (n). Bars show mean ± SEM. Source data are available online for this figure.

## Transcription suppresses D-loops independently of STR, Srs2, Mph1, and Rdh54

The Sgs1-Top3-Rmi1[BLM-TOPO3a-RMI1/2] (STR) helicase-topoisomerase complex, the Mph1[FANCM] and Srs2 helicases, and the Rdh54[RAD54B] dsDNA translocase suppress HR- and repeat-mediated gross chromosomal rearrangements (Putnam et al, 2009, 2016), disrupt or alter formation of D-loops in reconstituted in vitro reactions with Rad51 and Rad54 (Prakash et al, 2009; Fasching et al, 2015; Liu et al, 2017; Shah et al, 2020), and cause a reduction of the amount of D-loops detected in our system (Piazza et al, 2019; Xie et al, 2024; Hung et al, 2025). To address the genetic interactions between transcription and these *trans* D-loop disruption factors, we combined these individual mutations with a donor under the control of the silent *pDMC1* or the active *pTDH3* promoter. The two- to three-fold increase in D-loop levels detected in these mutants with the non-transcribed donor recapitulated previous findings obtained with the donors under the endogenous *pLYS2* promoter (Piazza et al, 2019) (Fig. 3D,E). None of them relieved the approximately tenfold inhibition imposed by transcription (Fig. 3D), indicating that transcription-mediated D-loop suppression does not require any of these factors. Moreover, the fold-change in these mutants over the wild-type background was overall similar with the silent and active promoters (Fig. 3E). Consequently, donor transcription is a distinct D-loop suppression pathway without detectable overlap with that conferred by these specialized HR regulators. Finally, the absolute fold-change in D-loop levels measured in these mutants (two- to three-fold) vs. that conferred by transcription (approximately sevenfold) shows

that transcription can be the main D-loop suppression pathway in cells, as a function of the local transcriptional activity.

The mismatch repair protein Msh2, also involved in heteroduplex rejection, was not implicated in D-loop reversal at our perfectly homologous substrates, irrespective of the transcriptional status of the donor (Fig. EV3B; no significant difference).

## The efficiency of D-loop suppression depends on transcription directionality

We addressed the impact of the directionality of transcription relative to that of the D-loop in two contexts: by repositioning the promoter at the donor site; or by inverting the region of homology near the break site (Fig. 4A). In these two independent "head-on" contexts, the RNA Pol II is set to encounter the 3′ junction of the D-loop rather than the 5′ junction (Fig. 4A).

Transcriptional activity at the donor was similar with promoters placed in the co-directional and head-on orientations, as verified by ChIP-qPCR against the Rpb1 subunit of RNA Pol II (Fig. EV4A). Donor transcription in the head-on orientation led to reduced D-loop levels, but the extent of this inhibition was less pronounced compared to the co-directional orientation (Fig. 4B,C). Specifically, D-loop levels were approximately threefold higher at donors transcribed in the head-on compared to the co-directional orientation (Fig. 4C). D-loops at the non-transcribed donor were not affected by promoter repositioning (Fig. 4B,C). Likewise, inversion of the homology region on the broken molecule partly alleviated the inhibition posed by donor transcription, while exerting no effect on D-loop levels at the non-transcribed donor

(Fig. 4D,E). Again, D-loop levels were approximately threefold higher with transcription in the head-on than in the co-directional orientation (Fig. 4E). These two independent sets of constructs show that transcription is not as effective at causing D-loop loss in the head-on than in the co-directional orientation. Consequently, a permissive orientation exists for D-loops to form in highly-transcribed genes.

### Donor transcription inhibits multi-invasion-induced rearrangements

Formation of a DNA break in the vicinity of a repeated sequence can elicit the formation of chromosomal rearrangements by a recombination process termed "multi-invasion recombination" (MIR). In MIR, the invasion of two independent donor molecules by a single repeat-containing Rad51-ssDNA filament leads to their translocation (Piazza et al, 2017; Reitz et al, 2023). In order to address whether transcription-dependent D-loop suppression could protect against MIR, we adapted a previously established genetic assay in which the DSB-inducible site, the internal donor, and the terminal donor are on different chromosomes (Piazza et al, 2017) (Fig. 5A). We placed the terminal donor under the control of inactive or low activity promoters (*pDMC1* and *pLYS2*) or highly active promoters (*pTDH3* and *pGAL1*) in our culture conditions (Fig. 5B). The internal promoter remained under the control of its native *pLYS2* promoter for selection purposes. Transcription of the terminal donor caused a 3.2- to up to 6-fold decrease in MIR frequency (Fig. 5B). In contrast, cells deficient for Sgs1, Mph1 and Srs2 caused MIR to increase by only ~1.2- to 3-fold in the absence of transcription (Fig. 5C). Hence, transcription of a single donor inhibits MIR more effectively than any of these *trans* HR regulators. These mutants caused similar fold increases in MIR whether or not the terminal donor was transcribed, corroborating genetically the independence between transcription and these HR regulators in promoting D-loop disruption determined molecularly (Fig. 3D,E). In conclusion, transcription inhibits the formation of MIR with efficiency that can exceed that conferred by conserved HR regulators involved in promoting genome maintenance.

### Donor transcription inhibits ectopic recombination

Formation of a DNA break within a repeated sequence can lead to the formation of chromosomal rearrangement by the canonical DSBR pathway (Szostak et al, 1983). In order to address whether donor transcription can inhibit canonical HR repair using an ectopic repeat, we adapted our DLC/DLE experimental system so that both DSB ends on chr. V share ~1 kb of homology to the donor on chr. II (Fig. 5D, Methods). These constructs enable repair completion and production of a gene conversion event that causes the loss of the HOcs and an EcoRI site (Fig. 5D). This gene conversion can either be associated with a non-crossover (NCO) or a crossover outcome (CO; which results in a balanced translocation between chr. II and V). Restriction sites located in-between and outside the homologous regions enabled quantifying formation of the NCO and CO repair products using a highly sensitive digestion/ligation procedure previously developed to quantify chromosomal rearrangements by qPCR (Fig. 5D) (Reitz et al, 2023). The break was efficiently repaired off the non-transcribed donor, with NCO and CO accounting for $50.4 \pm 5.0\%$ and $9.9 \pm 1.3\%$ of total broken

molecules 8 h post-DSB induction, respectively (Figs. 5E and EV5A). Donor transcription led to a ~6-fold decrease in repair efficiency, which equally affected NCO and CO ($7.9 \pm 1.7\%$ and $1.4 \pm 0.1\%$, respectively; Fig. 5E). No repair products were detected in the absence of a donor (Fig. 5E). Consistently, donor transcription inhibited cell survival approximately fivefold upon break formation, from $46.3 \pm 8.3\%$ with a non-transcribed donor to $10.3 \pm 2.6\%$ with a transcribed donor (Fig. 5F). These results show that transcription strongly suppresses both the canonical synthesis-dependent strand annealing and DSBR recombination pathways at an ectopic donor. Transcription is thus a major suppressor of repeat-mediated chromosomal rearrangements by inhibiting both DSBR and MIR (Fig. 5G).

### Donor transcription does not inhibit allelic recombination

Ectopic and allelic donors differ by the extent of homology available for repair. We addressed whether such extensive homologies could overcome the inhibitory effect imposed by transcription of a ~2 kb donor region corresponding to the DSB-flanking sites. To this end, we modified the inter-homolog hetero-allele repair system developed in diploid cells by the Symington laboratory (Ho et al, 2010). This system enables measuring the repair efficiency of an I-SceI-induced DNA break at the *ade2-I* locus onto the *ade2-n*-containing homologous chromosome and distinguishing the NCO, CO, and BIR repair outcomes by tracking the segregation of flanking *MET22*, *URA3*, *HPH*, and *NAT* markers (see Fig. 6A and Methods). The *ade2-n* donor was placed under the control of the inactive *pDMC1* promoter or the active *pTDH3* promoter (Figs. 6A and EV5B). Note that in this context, the transcriptional status of the region surrounding *ade2-n*, and that also shares homology with the broken chromosome, remains unchanged. The viability reached ~100% with both the *pDMC1* and the *pTDH3* constructs (Fig. 6B) without significantly affecting the distribution of NCO, CO and BIR repair outcomes (Fig. 6C). Consequently, the transcriptional status of the donor did not interfere with HR repair in an allelic context. It suggests that a highly-transcribed site in a broader non-transcribed region of homology is insufficient to exert the anti-recombination effect observed at ectopic sites.

## Discussion

Here, through direct D-loop detection in multiple genetic and mutant contexts, we show that donor transcription by RNA Pol II suppresses D-loops (and/or its precursor, paranemic Rad51-bound synaptic complex). Functionally, this layer of HR regulation promotes genome maintenance by inhibiting repeat-mediated chromosomal rearrangements. Hence, HR fidelity is not uniform along the genome, but instead depends on the local transcriptional activity.

### Putative mechanism of transcription-mediated D-loop suppression

D-loop suppression at a highly-transcribed donor occurs in *cis* and can be rapidly ( <10 min) switched on and off, strongly suggesting that D-loop suppression is a direct consequence of RNA Pol II

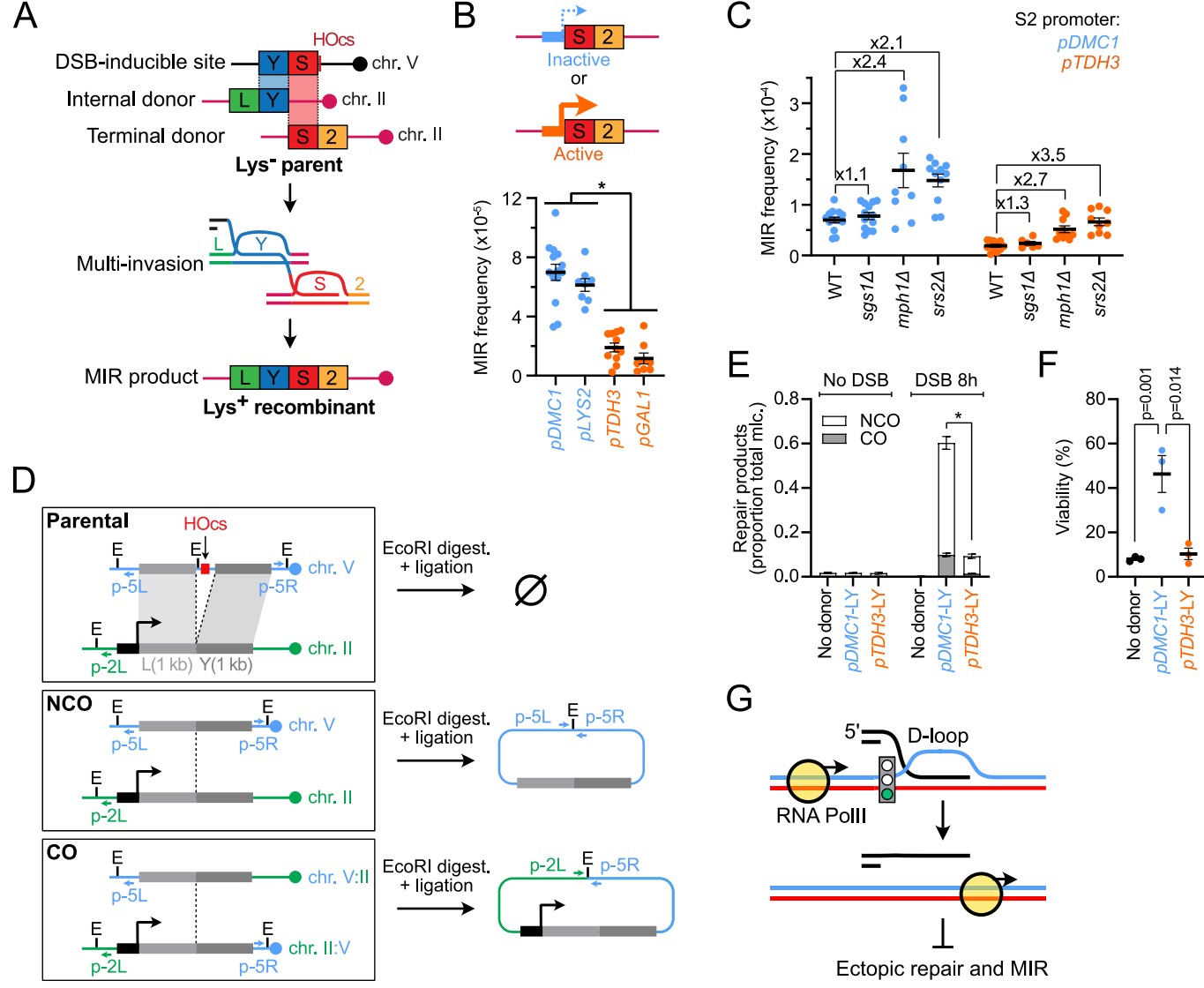

**Figure 5.  Donor transcription suppresses ectopic recombination and MIR.**

(**A**) Tripartite inter-chromosomal genetic system to study MIR. (**B**) Co-directional transcription of the terminal donor suppresses MIR (APY1130, APY1512, APY1077, and APY1075). *p < 0.0001 for all pairwise comparisons between inactive and active promoters (Mann–Whitney Wilcoxon test). (**C**) MIR with a transcriptionally inactive and active terminal donor in WT (APY1130 and APY1077), sgs1Δ (APY1693 and APY1692), mph1Δ (APY1682 and APY1681), and srs2Δ (APY1684 and APY1683) strains. (**D**) Inter-chromosomal ectopic recombination system and rationale of NCO and CO quantification by CR-capture (Reitz et al, 2023). "E" represents the position of EcoRI sites. (**E**) Transcription of an ectopic donor suppresses both NCO and CO repair product formation in WT cells (APY1188, APY1985, and APY1994). Data show mean ± SEM of n = 3 biological replicates. *denotes statistical significance for both NCO and CO (p = 0.0002 and 0.0003, respectively; Student t-test). (**F**) Transcription of an ectopic donor inhibits HR repair and causes loss of viability. P values were computed using a Student t-test. (**G**) Model for transcription-mediated suppression of ectopic recombination. (**B, C, F**) Data points show individual biological replicates (n). Bars show mean ± SEM. Source data are available online for this figure.

translocation at the donor. Supporting this suggestion, we could rule out secondary consequences of transcription, such as peripheral nuclear delocalization, RNA:DNA hybrids, and the RNA product acting in *trans* in D-loop suppression. It sets this mechanism of HR control apart from those implicating RNA molecules identified in yeast and human cell lines (Keskin et al, 2014; Meers et al, 2020; Ouyang et al, 2021; Liu et al, 2022).

RNA Pol II is a processive directional molecular motor threading along dsDNA, a process facilitated by TFIIS (Gnatt et al, 2001; Kettenberger et al, 2004; Charlet-Berguerand et al,

2006). Such active translocation may dissociate already formed D-loops by mechanically migrating its strand exchange junctions, an energetically neutral reaction. The fact that transcription mainly causes loss of co-directional D-loops (i) suggests that it does not solely act by preventing the upstream step of Rad51-ssDNA NPF binding to dsDNA, and (ii) makes it unlikely that D-loop dissociation is primarily mediated by topological changes at the donor. It instead suggests that the prioritization of transcription over the synaptic steps of HR depends on the type of DNA strand exchange junctions encountered by RNA Pol II, and/or on the

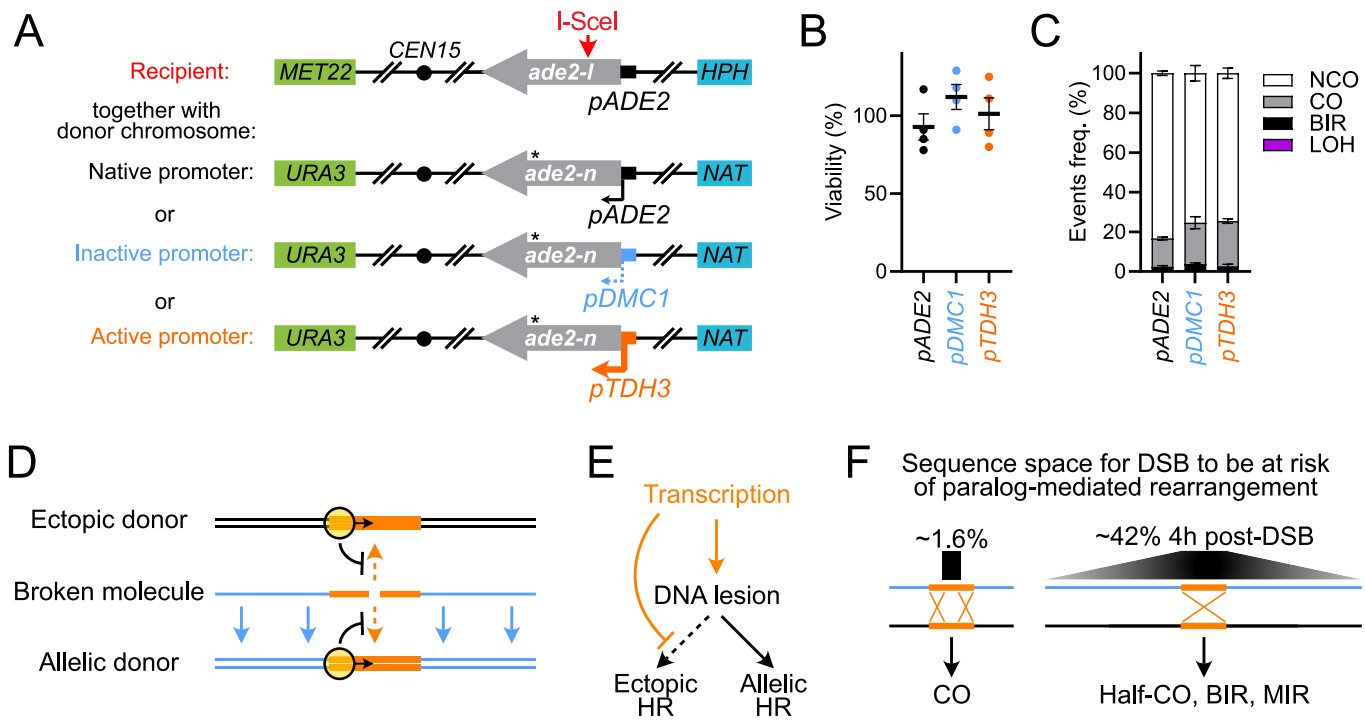

**Figure 6. A highly-transcribed gene does not suppress allelic HR.**

(A) Inter-chromosomal genetic system and *ade2-n* donor variants to study the efficiency and outcome of allelic HR (Ho et al, 2010). Diploids were made fresh by mating APY1899 (*ade2-I*) with APY1898 (*ade2-n*), APY1990 (*pDMC1-ade-n*), or APY1988 (*pTDH3-ade2-n*). (B) Viability following DSB induction in diploid strains bearing an *ade2-n* donor under control of the *pADE2* (APY1898 x APY1899), *pDMC1* (APY1990 x APY1899) or *pTHD3* promoter (APY1988 x APY1899). Points show individual biological replicates and mean ± SEM. No significant difference was detected. (C) Distribution of NCO, CO, BIR, and LOH events following DSB induction from colonies in (B) Data shows mean ± SEM of $n = 4$ biological replicates. No significant difference was detected (Welch *t*-test). (D) Donor transcription differently affects ectopic and allelic donors. (E) Transcription stimulates the formation of recombinogenic lesions and biases their repair towards allelic donors. (F) DSB localization relative to a repeat susceptible to causing a chromosomal rearrangement upon CO (left), or half-CO, BIR and MIR (right). In the latter cases, the sequence space increases with the resection tract length. The proportion of the genome involved is indicated. Source data are available online for this figure.

presence of HR proteins decorating them relative to an incoming RNA Pol II. Consistently, RNA Pol II obtained from HeLa cells extract could traverse a model co-linear D-loop substrate in vitro, while a head-on D-loop was a roadblock (Pipathsouk et al, 2017). The precise structural basis for this orientation-dependent behavior, at the heart of the prioritization between the two processes, remains to be determined.

## Role of transcription-recombination prioritization in genome maintenance

Transcription has primarily been recognized as a pro-recombinogenic process through the formation of replication-dependent DNA lesions in *S. cerevisiae* (Keil and Roeder, 1984; Thomas and Rothstein, 1989; Deshpande and Newlon, 1996; Saxe et al, 2000; Prado and Aguilera, 2005), a phenomenon largely conserved (reviewed in (Browning and Merrikh, 2024)). Early work from the Jinks-Robertson laboratory using a spontaneous ectopic recombination selection system suggested an additional anti-recombination function of transcription, presumably acting downstream of transcription-induced lesion formation (Saxe et al, 2000). Here, using site-specific DSB induction systems, we established that donor transcription inhibited ectopic repair and the formation of repeat-mediated chromosomal rearrangements by both CO and MIR. The magnitude of the suppression of ectopic recombination (~6-fold, Fig. 5D) and MIR (~3.5-fold, Fig. 5B) was commensurate with that measured at the D-loop joint molecule levels with donors of the same length (~7-fold and ~3-fold, respectively, Fig. 1C,E). Consequently, the prevention of ectopic recombination upon donor transcription predominantly, if not exclusively, arises as a consequence of D-loop suppression (Fig. 5G), which rules out additional roles at downstream intermediates and steps of the HR pathway.

The presence of a highly-transcribed gene in a broader allelic context did not inhibit HR, consistent with its localized, *cis* nature. The extensive homologies specific to allelic donors presumably allow for DNA strand invasion to occur, and the repair to proceed, from regions of homology other than the transcribed one (Fig. 6D), even if localized at a distance from the region immediately flanking the break site (Inbar and Kupiec, 1999). Furthermore, homology length (Fig. 1E) and spatial proximity of the donor (Fig. 2C), two hallmarks of the sister chromatid, partly counteracted the effect of transcription. Consequently, transcription is not a general anti-recombination mechanism, but an anti-ectopic recombination mechanism (Fig. 6E).

The budding yeast genome contains ~7% or repeated sequences that can mediate HR-dependent chromosomal rearrangements (Richard et al, 2008). Among these, paralogous gene families

involved in dosage amplification are highly-transcribed and retain high (>70%) sequence similarity (~270 genes, Fig. EV6A; Dataset EV2) (Kuzmin et al, 2022). As such, they both generate recombinogenic lesions and provide substrates for ectopic repair, a combination expected to synergize for genome destabilization. We propose that transcription-induced D-loop suppression specifically undermines this synergism (Fig. 6E).

The inhibition of both CO and MIR shows that transcription can inhibits HR in contexts in which a DSB occurs within, or in the vicinity of a repeated sequence, respectively (Savocco and Piazza, 2021) (Fig. 6F). Consequently, although highly similar paralogous genes cover only ~3% of the genome, the fraction of repair events benefiting from transcription-mediated D-loop suppression is likely to be much greater. In the canonical DSBR model, formation of a chromosomal rearrangement at these genes upon CO formation requires a DSB to fall squarely within the repeat, >300 bp away from the edges (Inbar et al, 2000). In this scenario, only ~1.6% of randomly distributed DSBs would be at risk of causing a chromosomal rearrangement between paralogs (Fig. 6F). However, the possibility to undergo half-CO, BIR, or MIR from internal regions of homologies from a single resected end greatly expands the sequence space within which a DSB can induce a chromosomal rearrangement: the repeat only needs to be exposed by resection (Fig. 6F, and see (Piazza and Heyer, 2019; Reitz et al, 2023)). In the most conservative case of random DSB distribution and assuming a resection speed of ~4 kb/hr (Zhu et al, 2008; Mimitou and Symington, 2008), the sequence space for paralog-mediated rearrangements expands to 15% and up to 42% of the genome at 1 and 4 h after DSB formation, respectively (Fig. EV6B). These are likely conservative figures, as the distribution of endogenous recombinogenic lesions is not random, but enriched around these highly-transcribed genes. Consequently, the transcription-mediated suppression of recombination at highly-transcribed paralogous genes likely plays a major role in maintaining genome structure in budding yeast.

# Methods

### Reagents and tools table

| Reagent/resource | Reference or source | Identifier or catalog number |
| --- | --- | --- |
| **Experimental models** | | |
| W303 RAD5+ S. cerevisiae strains | This study | Table EV1 |
| **Recombinant DNA** | | |
| trp1::GAL-HO | Pannunzio et al, DNA Rep 2008 | https://doi.org/10.1016/j.dnarep.2008.02.003 |
| ura3::LY-HOcs | Piazza et al, Cell 2017 | Dataset EV1 |
| ura3::L-HOcs | This study | Dataset EV1 |
| ura3::L(reverse)-HOcs | This study | Dataset EV1 |
| ura3::L-HOcs-Y | This study | Dataset EV1 |
| lys2::LY | Piazza et al, Cell 2017 | Dataset EV1 |

| Reagent/resource | Reference or source | Identifier or catalog number |
| --- | --- | --- |
| lys2::pDMC1-LY | This study | Dataset EV1 |
| lys2::pTDH3-LY | This study | Dataset EV1 |
| can1::LY | Piazza et al, NCB 2021 | Dataset EV1 |
| can1::pDMC1-LY | This study | Dataset EV1 |
| can1::pTDH3-LY | This study | Dataset EV1 |
| lys2::L | This study | Dataset EV1 |
| lys2::pDMC1-L | This study | Dataset EV1 |
| lys2::pTDH3-L | This study | Dataset EV1 |
| lys2::L(reverse) | This study | Dataset EV1 |
| lys2::pDMC1-L(reverse) | This study | Dataset EV1 |
| lys2::pTDH3-L(reverse) | This study | Dataset EV1 |
| lys2::pGAL1-L | This study | Dataset EV1 |
| lys2::pCUP1-L | This study | Dataset EV1 |
| lys2::pTetO2-L | This study | Dataset EV1 |
| lys2::S2 | Piazza et al, Cell 2017 | Dataset EV1 |
| lys2::pDMC1-S2 | This study | Dataset EV1 |
| lys2::pTDH3-S2 | This study | Dataset EV1 |
| lys2::pGAL1-S2 | This study | Dataset EV1 |
| ade2-I | Ho et al, Mol Cell 2010 | https://doi.org/10.1016/j.molcel.2010.11.016 |
| ade2-n | Ho et al, Mol Cell 2010 | Dataset EV1 |
| pDMC1-ade2-n | This study | Dataset EV1 |
| pTDH3-ade-n | This study | Dataset EV1 |
| pYES2(empty) | Thermofisher Sci. | Dataset EV1; Cat# V825120 |
| pYES2-RNH1 | This study | Dataset EV1 |
| **Antibodies** | | |
| Mouse anti-Rpb1 clone CTD4H8 | Sigma-Aldrich | Cat#05-623 |
| **Oligonucleotides and other sequence-based reagents** | | |
| PCR primers | This study | Table EV3 |
| **Chemicals, enzymes and other reagents** | | |
| Zymolyaze 100 T | Carl Roth | Cat#9329 |
| Formaldehyde | Sigma-Aldrich | Cat#F8775 |
| EcoRI-HF | NEB | Cat#R3101 |
| HindIII-HF | NEB | Cat#R3104 |
| T4 DNA ligase | NEB | Cat#M0202 |
| Proteinase K | NEB | Cat# P8107S |
| RNAseA | EUROMEDEX | Cat# 9707-C |
| SSO Advanced Universal SYBR supermix | Bio-Rad | Cat#1725274 |
| iTaq™ Universal SYBR® Green One-Step Kit | Bio-Rad | Cat#1725150 |

| Reagent/resource | Reference or source | Identifier or catalog number |
|---|---|---|
| Qubit DNA high-sensitivity kit | Thermo Fisher Scientific | Cat#Q32851 |
| Nucleospin RNA kit | Machery Nagel | Cat# MN06 740588.250 |
| Dynabeads proteins G | Thermo Fisher Scientific | Cat#10003 |
| Sodium DL-lactate | Sigma-Aldrich | Cat# L1375 |
| Doxycycline | Sigma-Aldrich | Cat# D9891 |
| Trioxsalen | Sigma-Aldrich | Cat# T6137 |
| **Software** | | |
| Prism 10 | Graphpad | https://www.graphpad.com/ |
| CFX Maestro 2.0 | Bio-Rad | |
| Biopython 1.78 | Cock et al, Bioinformatics 2009 | https://biopython.org/ https://doi.org/10.1093/bioinformatics/btp163 |
| Seaborn 0.12.2 | Waskom JOSS 2021 | https://seaborn.pydata.org/ https://doi.org/10.21105/joss.03021. |
| Jupyter notebook 6.5.7 | JupyterLab | https://jupyter.org/ |
| **Other** | | |
| CFX96 Touch Deep Well Real-Time PCR Detection System | Bio-Rad | Cat#3600037 |

## *Saccharomyces cerevisiae* strains and genetic constructs

The haploid and diploid *Saccharomyces cerevisiae* strains used in this study derive from the W303 *RAD5+* background. The genotypes are provided in Table EV1. The annotated sequences of all the DSB-inducible and donor constructs are provided in Dataset EV1.

The *HO* gene was placed under the control of the *pGAL1/10* promoter at the *TRP1* locus on chromosome IV (Pannunzio et al, 2008), the point mutation at the HO cut-site (*HOcs*) present at the mating-type locus (*MAT*) on chromosome III to prevent its cleavage by HO (*MAT*a-inc and *MAT*α-inc), and the DSB-inducible construct have been described previously (Piazza et al, 2019, 2018; Reitz et al, 2022). Briefly, the DSB-inducible construct consists of the *HOcs* introduced in place of the *URA3* gene on chromosome V (−16 to +855 from the start codon). The *HOcs* is flanked on its left side by a 1- or 2- kb-long fragment of the 5' end of the *LYS2* gene, and are referred to as "*L*" (+3 to +1042 bp from start codon) and "*LY*" (+3 to +2087 bp from start codon), respectively. In the *L*-inverted construct, the orientation of the "*L*" sequence is flipped but remains on the left side of the *HOcs*. A 453 bp sequence containing 327 bp of the phage PhiX174 genome (coordinates 449 to 775) flanked by multiple restriction sites (including the EcoRI site and the qPCR primer used in the DLC assay) is present upstream of the "*L*" sequence (Piazza et al, 2019). The "*L*" and "*LY*" donors are located at the endogenous *LYS2* locus on chr. II (the "*YS2*" or "*S2*"

remainder of the gene has been removed). The lack of homology between the donor and the right side of the break prevents repair by synthesis-dependent strand annealing or DSBR (Pâques and Haber, 1999; Szostak et al, 1983). The orientation of the donor towards *CEN2* prevents repair completion by BIR (Morrow et al, 1997; Pham et al, 2021). This system thus precludes repair completion, so as to retain a constant number of cells undergoing repair at all time points (Reitz et al, 2022).

In the co-directional orientation, the "*L*" and "*LY*" donors are under the control of the native *pLYS2* promoter (−155 to −1 bp from the start codon), the *pDMC1*, the *pTDH3*, the *pGAL1*, or the *pCUP1* promoters. The S288c coordinates of the promoters are listed in Table EV2. The *pTetO2-TATA* promoter was obtained from the pST1873 vector. In the head-on orientation, the promoters and the *tLYS2* terminator (+1 to +70 bp from the stop codon) were exchanged and reverse-complemented. The artificial *tGuo1* terminator (5′-TATATAACTGTCTAGAAATAAAGGTG-CAGGCATTTCAAA-3′) (Curran et al, 2015) was further added downstream of *tLYS2*.

Strains with a second donor, which is inserted 85 kb away from the DSB site at the *CAN1* locus on chromosome V, are used to compare the D-loop formation at the intra-chromosomal and inter-chromosomal donors (Piazza et al, 2021).

In the ectopic repair system, the DSB-inducible construct consists of the HOcs flanked on the left by the "*L*" sequence and on the right by the "*Y*" sequence, both ~1-kb-long. An EcoRI site was introduced between the "*L*" sequence and the HOcs, and a 403 bp-long random sequence was added together with an EcoRI site downstream of the "*Y*" sequence. This additional sequence and site enable quantifying NCO and CO formed upon ectopic repair at the "*LY*" donor. The rest of the construct is identical to the one used for the DLC assay.

The diploid strains for the MIR study were as in (Piazza et al, 2017). They contain a heterozygous DSB-inducible construct at *ura3* on chromosome V. The DSB-inducible construct consists of the HOcs flanked by the central, 2-kb-long part of the *LYS2* gene (YS). The donor in these strains is divided into two parts on each copy of chromosome II; the first copy carries the first half of the *LYS2* gene (LY), and the second copy carries the second half of the *LYS2* gene (S2), without sequence overlap.

Regarding the inter-homolog *ade2* hetero-allele repair system (Ho et al, 2010), the *pADE2* promoter (−1 to −525 bp from the start codon) in front of the donor *ade2-n* allele was replaced by the *pDMC1* or *pTDH3* promoter.

The *RNAseH1* (*RNH1*) was overexpressed from the *pGAL1* promoter on the *URA3*-containing 2μ plasmid pYES2 (Thermo Fisher cat. V82520). The empty pYES2 plasmid was used as a negative control. Clones exhibiting equivalent copy number (~20) of the plasmid were selected by qPCR against the *URA3* gene using primers 5′-TACAGTCAAATTGCAGTACTC and 5′-CTGCTAACATCAAAAGGCCTC.

## Culture media

Culture media were prepared according to standard protocols (Treco and Lundblad, 2001). Yeast Extract Peptone media (YP) are composed of 1% bacterial and yeast extract and 2% peptone, with variations in the carbon source: 2% dextrose (YPD), 2% lactate (YPL), or 2% galactose (YPGal). The synthetic medium contained

0.17% yeast nitrogen base, 0.5% ammonium sulfate, 2% dextrose, and 0.2% all amino acids for the synthetic complete (SC) or 0.2% appropriate amino acids dropout for the synthetic selective media (SD-AA). The synthetic dropout galactose contained 2% galactose. Additionally, the solid media contained 2% agarose.

Note that the water used in media preparation has changed over the course of this study from osmosis water to ultrapure water obtained after Milli-Q filtering, which improved yeast growth and led to slight changes in D-loop levels. Consequently, D-loop levels were always compared between matched samples grown in the same media and processed in parallel.

## Induction of DSB and cell treatments

*HO* expression was induced upon the addition of 2% galactose to YPL cultures reaching OD600 ~0.5.

In the case of *pCUP1* activation with copper, cultures were grown in SC-Lactate media up to OD600 ~0.5, and *HO* expression was induced upon addition of 2% galactose. Copper sulfate salt ($CuSO_4$ in water) was added at 5, 20, or 50 µM 110 min post-DSB induction, corresponding to 5 to 10 min prior to DLC sample collection.

In the case of RNAseH1 overexpression and its matched empty vector control, saturated precultures prepared in 2% glucose-containing SD-Uracil media were diluted in lactate-containing synthetic dropout lacking uracil and grown overnight to OD600 ~0.5. *HO ± RNH1* overexpression was induced upon addition of 2% galactose.

In the case of transcriptional control of the donor by TetR-Ssn6 and TetR-VP16, exponential cultures in SC-Lactate reaching OD600 ~0.5 were split, and 20 µg/mL doxycycline (Sigma-Aldrich cat. D9891) or the equivalent ethanol concentration was added together with 2% galactose.

## D-loop capture (DLC) assay

The DLC was performed as in (Piazza et al, 2019; Reitz et al, 2022), including the psoralen crosslink reversal step described in (Reitz et al, 2022). Briefly, a site-specific DSB was induced upon overexpression of the HO endonuclease by adding galactose at a final concentration of 2% to an exponentially growing cell culture in YEP-lactate media. About $2 \times 10^8$ cells were collected prior to, and at various time points after, galactose addition. For DLC, cells were resuspended in crosslinking solution (0.1 mg/mL Trioxsalen (Sigma-Aldrich T6137), 50 mM tris-HCl pH 8.0, 50 mM EDTA, 20% ethanol) and the DNA was crosslinked with ~32 J/cm² UV-A (365 nm) irradiation in a Bio-link – BLX365 (Vilber-Lourmat, cat. 611110831) with permanent orbital agitation (~50 rpm). Cell were spheroplasted for 15 min at 37 °C with 3.5 µg/mL Zymolyase 100T in spheroplasting buffer (0.4 M sorbitol, 0.4 M KCl, 40 mM phosphate buffer pH 7.4, and 0.5 mM $MgCl_2$) and washed twice with spheroplasting buffer and three times with 1X Cutsmart buffer (20 mM tris acetate pH 7.9, 50 mM potassium acetate, 10 mM magnesium acetate, 100 µg/mL BSA) at 4 °C. Pellets were resuspended in 1.4X Cutsmart buffer, flash frozen in liquid nitrogen and stored at −70 °C. Cells were lysed upon addition of 0.1% SDS at 65 °C for 10–15 min in the presence of an 80mer oligonucleotide (APO563; Table EV3), whose annealing restores the EcoRI restriction site on the resected broken molecule. DNA

was recovered from the spheroplasts, digested by EcoRI-HF (NEB, cat. R3101L), and ligated with T4 ligase (NEB, cat. M0202) at low concentration (~$1.8 \times 10^4$ genomes/µl). DNA was extracted with phenol-chloroform after protein degradation using proteinase K. Psoralen inter-strand crosslinks and adducts was reversed in 100 mM KOH at 90 °C for 30 min. The pH was neutralized upon addition of 66 mM of NaOAc, pH 5.2. Approximately $6 \times 10^5$ genome equivalents were used per quantitative PCR (qPCR) reaction, performed in duplicate, on a CFX96 Touch Deep Well Real-Time PCR Detection System (Bio-Rad cat. 3600037), using the SSO Advanced Universal SYBR Green Supermix (Bio-Rad, cat. 1725274), following manufacturer's instructions. Primers used are listed in Table EV3. qPCR analysis were performed as described in (Reitz et al, 2022) using Bio-Rad CFX Maestro and Microsoft Excel.

## D-loop extension (DLE) assay

The DLE was performed as in (Piazza et al, 2018; Reitz et al, 2022). Samples were collected and processed as for the DLC assay, except that no psoralen crosslinking was performed. At the lysis step, HindIII restriction sites were restored upstream of the region of homology on the broken molecule and downstream of the donor site upon annealing of APO581 and APO640, respectively (Table EV3). DNA was digested with HindIII-HF (NEB, cat. R3104L) instead of EcoRI. Quantitative PCR was performed using primers listed in Table EV3 and data analyzed as described in (Reitz et al, 2022).

## MIR translocation assay

The MIR translocation assay was performed using diploid *S. cerevisiae* strains as described in (Piazza et al, 2017), except that the DSB was not induced upon galactose addition in liquid media, but upon direct plating of an exponential culture in liquid YEP-lactate media on galactose-containing plates. Cells were spread on YPGal and YPD to determine viability, and on SDGal-Lysine and SDGlu-Lysine to determine the frequency of Lys+ recombinants in the presence and absence of DSB, respectively. The viability and Lys$^+$ frequencies were determined by counting colonies after incubating the plates for 2–3 days at 30 °C.

## Ectopic recombination assay

The same DSB induction procedure was performed as for the DLC assay. Samples were collected and processed as in the CR-Capture assay described in (Reitz et al, 2023). Briefly, $5 \times 10^8$ cells were collected prior to, and 8 h after, DSB induction. Cells were spheroplasted with Zymolyase and the genomic DNA extracted with a standard phenol-chloroform-isoamyl alcohol (25:24:1) procedure followed by RNA digestion with RNAseA. DNA was quantified on a Qubit using the HS Assay Kit. 500 ng of DNA was digested with EcoRI-HF (NEB cat. R3101L). About 80 ng of digested DNA was used for a ligation reaction using the T4 ligase (NEB, cat M0202) in dilute conditions (~$10^4$ genomes/µl). The ligated DNA was then purified by phenol-chloroform-isoamyl alcohol (25:24:1) extraction, precipitated with ethanol and resuspended in 10 mM Tris-HCl 0.5 mM EDTA. The chimeric junctions produced upon ligation of NCO and CO molecules, a loading control, as well as a 2-kb-long circularization control (*DAP2*), were

quantified by qPCR using the primers listed in Table EV3. Approximately $10^5$ haploid genomes equivalent was used per qPCR reaction. The amount of NCO and CO circles were normalized onto the circularization efficiency to obtain the frequency of NCO and CO products.

Cell viability following DSB induction was determined by plating an exponentially growing cell culture in YEP-lactate on YPD (No DSB) and YPGal (DSB induction) plates and counting colonies after 4 days at 30 °C.

## Allelic recombination assay

Diploid strains were freshly constructed before each time course by crossing the appropriate haploid strains, and individual clones were selected upon streaking on Nourseothricin- and Hygromycin B-containing YPD media. DSB induction was performed as in (Ho et al, 2010), except that cells were grown in YEP-lactate instead of YEP-raffinose media. Briefly, DSB was induced upon addition of 2% galactose to an exponentially growing culture in YEP-lactate media, and the cells were plated on YPD 2 h after induction. The plates were incubated at 30 °C for 2–4 days and replica-plated on YPD + HYG, YPD + NAT, YPD + HYG + NAT, SC-Met, SC-Ura, and SC-Ura-Met to determine the segregation of the *HYG*, *NAT*, *MET22*, and *URA3* parental markers and infer NCO, CO, BIR, and chromosome loss events (Ho et al, 2010). Cells were also replica-plated on SC-Ade+Gal media to eliminate red colonies that had not experienced a DSB during the 2 h induction period.

Note that the replacement of the *pADE2* promoter on the donor introduces a short region of heterology between the broken and the donor chromosomes, preventing straightforward comparison with the unmodified strain (i.e., bearing the *ade2-n* allele under control of its endogenous *pADE2* promoter; Fig. 6A). Furthermore, the red/white colony analysis that reports on long vs. short-tract gene conversion in this system could not be performed as it requires *ADE2* expression (Ho et al, 2010), a precondition not fulfilled when under the control of the meiosis-specific *pDMC1* promoter.

## RNA extraction and RT-qPCR

RNA extraction was performed using the Nucleospin RNA kit (Machery Nagel, ref: MN06 740588.250) following the manufacturer's instructions. The quantitative reverse transcription PCR (RT-qPCR) was performed on a CFX96 Real-Time System (Bio-Rad), using an iTaq™ Universal SYBR® Green One-Step Kit (Bio-Rad, cat. 1725150) following the manufacturer's instructions. Primers used are listed in Table EV3.

## Quantitative chromatin immunoprecipitation (ChIP-qPCR)

Approximately $1.5 \times 10^8$ cells were crosslinked 2 h post-DSB induction in 3% formaldehyde. Cell lysis was performed in 300 μl of lysis solution (50 mM HEPES-KOH pH8, 140 mM NaCl, 1 mM EDTA pH 8, 1% Triton X-100, 0.1% sodium deoxycholate, and 1 mM PMSF supplemented with a protease inhibitor cocktail (Roche cat. 11836170001)) with ø 500 μm acid-washed beads using a Precellys (6800 rpm for 12 s, rest in ice for 45 s, repeated four times). The chromatin was sheared by sonication using a Covaris (240 W peak power, 20% duty factor and 200 cycles for 10 min). The lysate was clarified by centrifugation at $10,000 \times g$ at 4 °C.

RNA Pol II immunoprecipitation was performed using a mouse anti-RNA polymerase II antibody (anti-Rpb1 clone CTD4H8, Sigma-Aldrich cat. 05-623) overnight followed by 2 h incubation with equilibrated Dynabeads proteins G (Thermo Fisher cat. 10003) at 4 °C. Beads were washed three times in lysis solution, three times in lysis solution supplemented with high salt (500 mM NaCl), twice with a wash solution (10 mM Tris-HCl, 500 mM LiCl, 1 mM EDTA pH 8, 0.5% Igepal CA-630, 0.1% sodium deoxycholate, and 1 mM PMSF), and once with the TE-Na buffer (10 mM Tris-HCl pH 8, 1 mM EDTA, and 50 mM NaCl). Elution was performed using 1% SDS at 65 °C, and the crosslink was reversed with 3% SDS overnight at 65 °C. Following RNA degradation using RNase A and protein degradation using proteinase K, DNA was purified upon phenol-chloroform extraction and ethanol precipitation and resuspended in TE buffer (10 mM Tris-HCl, pH 8, and 1 mM EDTA). DNA was quantified using a Qubit dsDNA HS assay kit (Thermo Fisher cat. Q32854). RNA Pol II occupancy of the donor was analyzed by quantitative PCR using primers targeting the 3′ and 5′ ends of the donor (Table EV3). The IP over input ratio was normalized over that at the *ARG4* gene.

## Statistical analysis

Most statistical comparisons were performed using a non-parametric two-tailed Mann–Whitney–Wilcoxon test. When $n < 4$, a two-tailed Student *t*-test was used, making the assumption of a normal data distribution. Statistical tests were performed with GraphPad Prism 10. The statistical significance α cutoff was set at 0.05. Data were paired whenever appropriate. The test used is indicated in the figure legends.

In most instances, biological replicates (n) are represented as individual points. Bars and error show mean ± SEM. The representations used are indicated in the figure legends.

No blinding was conducted.

## Paralogous genes analysis

A list of paralogous gene families was manually assembled upon review of the literature. From that list, the sequence similarity between each paralogous genes was computed with Python using the "Script_paralogs_similarity.ipynb" notebook. Briefly, each gene sequence was retrieved from NCBI RefSeq and pairwise sequence comparison was performed with the pairwise2.align.globalxx module of Biopython with default parameters. Scores <0.7 were removed, and the matrix was plotted with Matplotlib and Seaborn.

The coordinates for these genes were retrieved from https://www.alliancegenome.org. The genomic intervals susceptible to expose >300 bp of at least one paralogous gene as a function of DSB location and resection tract length was computed with Python by extending their genomic intervals using the "Script_paralogs_resection.ipynb" notebook. Overlapping intervals were merged and the total genome fraction computed.

The scripts have been generated with the assistance of ChatGPT.

# Data availability

Scripts and input data: Github (https://github.com/Piazzalab/Djeghmoum2025). No other datasets have been deposited in external repositories.

The source data of this paper are collected in the following database record: biostudies:S-SCDT-10_1038-S44318-025-00541-x.

## Peer review information

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

## Acknowledgements

We thank the Piazza, Bernard, and Jost lab members, as well as Wolf-Dietrich Heyer, Bertrand Llorente, and Laurent Duret for stimulating discussions and helpful suggestions. We are grateful to Vincent Vanoosthuyse for stimulating discussions and his critical reading of the manuscript, and Lorraine Symington for sharing plasmids and strains. We are also grateful to Gérard Mazon and Diedre Reitz for their advice regarding the *ade2* recombination assay. This research was supported by the European Research Council (ERC) under the European Union's Horizon 2020 to AP (ERC grant agreement 851006) and a 4th-year PhD fellowship extension from the Fondation pour la Recherche Médicale to YD (FDT202404018451).

## Author contributions

**Yasmina Djeghmoum**: Data curation; Formal analysis; Investigation; Visualization; Writing—original draft. **Aurèle Piazza**: Conceptualization; Formal analysis; Supervision; Funding acquisition; Validation; Investigation; Visualization; Writing—original draft; Project administration; Writing—review and editing.

   Source data underlying figure panels in this paper may have individual authorship assigned. Where available, figure panel/source data authorship is listed in the following database record: biostudies:S-SCDT-10_1038-S44318-025-00541-x.

## Disclosure and competing interests statement

The authors declare no competing interests.

# Expanded View Figures

**Figure EV1.  Co-directional donor transcription suppresses nascent D-loops (related to Fig. 1).**

(A) Rationale of the proximity ligation-based D-loop Capture (DLC) and D-loop Extension (DLE) assays. (B) ChIP-qPCR of the Rpb1 subunit of RNA Pol II at the 5′ and 3′ end of the donor 2 h post-DSB induction. (C) RT-qPCR of the RNA produced at the donor and at the downstream *RAD16* gene 2 h post-DSB induction. (D) Multiple controls of the DLC experiments with varying transcriptional levels at the donor site. Left: Quantification of the DSB frequency at *HOcs* 2 h post-DSB induction. Middle: control of the EcoRI digestion efficiency in the broken and donor molecules. Right: Circularization efficiency of a *GAL3*-containing fragment as a function of its transcriptional activity. (E) Correlation between the D-loop levels and the RNA Pol II amount at the donor (left) and donor transcript (right). Data show mean ± SEM. From data in (B) (Pol II IP), (C) (RT-qPCR), and Fig. 1C (DLC). (F) RT-qPCR of the RNA produced at the 1-kb- and 2-kb-long donors 2 h post-DSB induction. (B–D, F) Data points show individual biological replicates (*n*). Bars show mean ± SEM.

▶

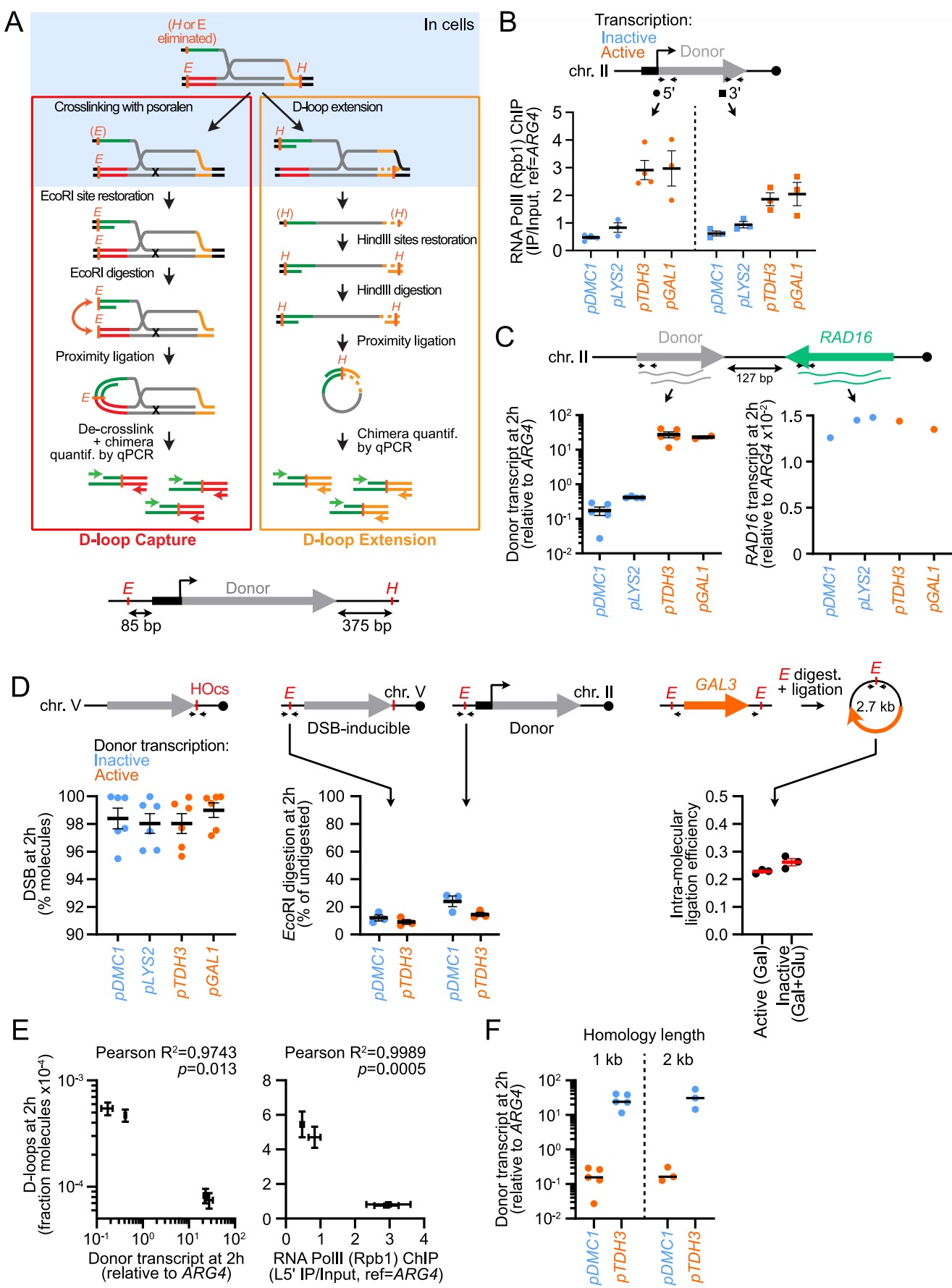

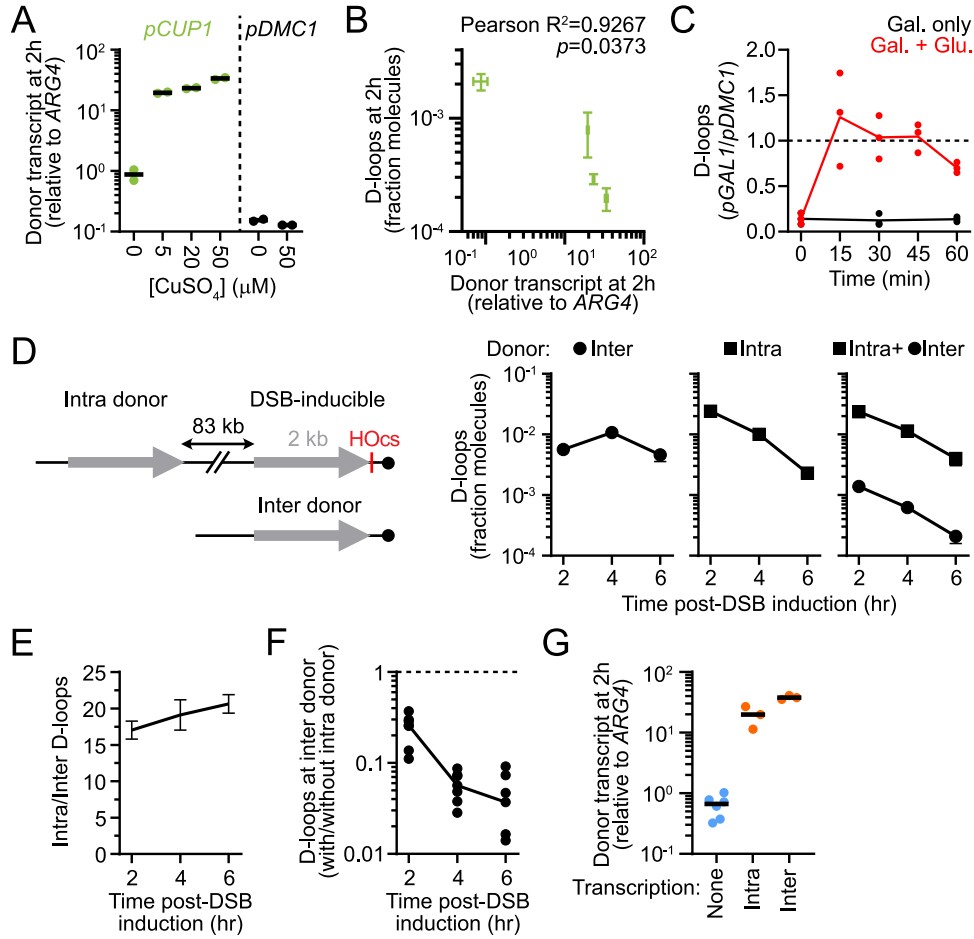

**Figure EV2. Donor transcription suppresses D-loops in *cis* (related to Fig. 2).**

(**A**) RT-qPCR of the RNA produced at the donor with varying copper concentrations 2 h post-DSB induction. (**B**) Correlation between the D-loop levels and donor transcript levels with varying copper concentrations. Data show mean ± SEM. From data in (**A**) (RT-qPCR) and Fig. 2A (DLC). (**C**) Ratio of D-loops formed at the donor site under the control of the *pGAL1* over the *pDMC1* promoter in contexts in which the *pGAL1* promoter is active (Gal. only) or at increasing time post-shutoff (Gal. + Glu.). From data in Fig. 2B. (**D**) Kinetics of D-loop levels at an intra-chromosomal and/or an inter-chromosomal donor (APY266, APY826, and APY809). Data show mean ± SEM of $n \geq 7$ (inter donor only), $n = 1$ (intra donor only), and $n \geq 7$ (intra+inter donor) biological replicates. (**E**) Ratio of D-loops formed at the intra donor over the inter donor when both are present (APY809). From data in (**C**). Data show mean ± SEM of $n \geq 7$. (**F**) Fold decrease of the D-loop at the inter donor when the intra donor is present. From data in (**D**) acquired in parallel ($n = 6$). (**G**) RT-qPCR of the donor RNA produced in strains bearing an intra and an inter donor either non-transcribed (APY809), with the intra donor transcribed (APY1587), or with the inter donor transcribed (APY1709). (**A, C, F, G**) Data points show individual biological replicates ($n$). Bars show mean ± SEM.

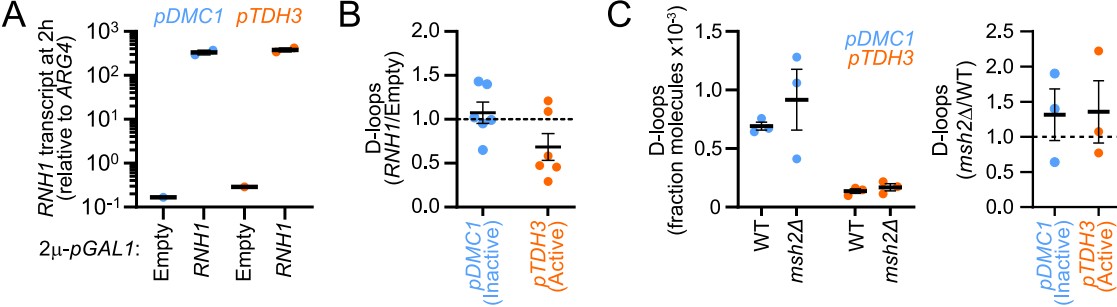

**Figure EV3.   Genetic determinants of transcription-mediated D-loop suppression (related to Fig. 3).**

(**A**) *RNH1* transcript levels determined by RT-qPCR as a function of the overexpression vector 2 h post-DSB induction. (**B**) D-loops level fold change at transcriptionally inactive and active donors upon *RNaseH1* overexpression. From data in Fig. 3A. (**C**) Left: D-loop levels at transcriptionally inactive and active donors in a WT (APY502 and APY725) and a *msh2Δ* (APY1608 and APY1610) strain. Right: relative mutant values compared to a WT strain assayed in parallel. No statistically significant differences were detected between *msh2Δ* and WT cells. (**A–C**) Data points show individual biological replicates (*n*). Bars show mean ± SEM.

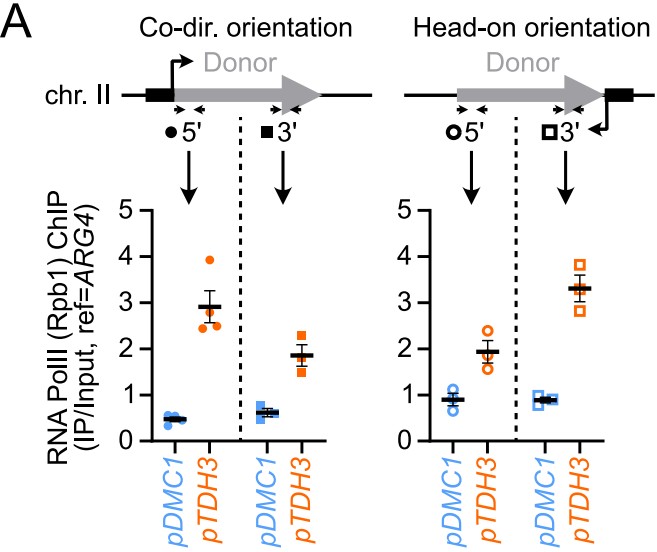

**Figure EV4. Effect of transcription directionality on RNA Pol II enrichment at the donor (related to Fig. 4).**

(A) ChIP-qPCR of the Rpb1 subunit of RNA Pol II at the 5′ and 3′ end of the donor 2 h post-DSB induction with promoters in the co-directional or head-on orientation.

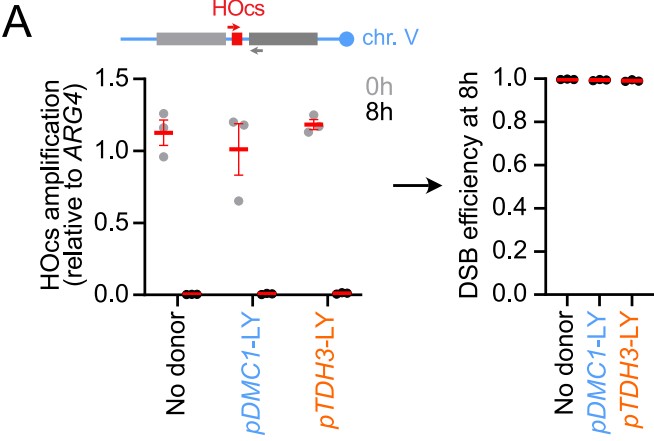

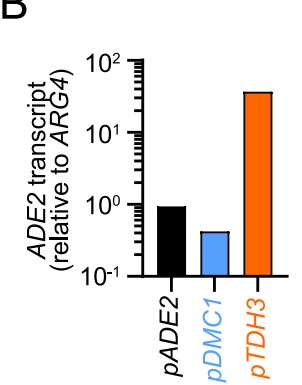

**Figure EV5.   Controls for the ectopic and allelic recombination systems.**

(**A**) Efficiency of DSB formation in the ectopic repair system. Left: Amplification across the HOcs prior to and 8 h post-DSB induction. Right: Deduced DSB efficiency at 8 h post-DSB induction. Data points show individual biological replicates (*n*). Bars show mean ± SEM. (**B**) Abundance of the *ADE2* transcript in diploid cells bearing the *ade2-n* donor under control of different promoters, scored after 2 h of DSB induction and 1 h of induction shutoff upon glucose addition in liquid media. This condition best matches the context in which DSB repair is expected to take place upon plating cells on YPD media following I-SceI induction for 2 h in liquid media. Data shows a single biological replicate (*n* = 1).

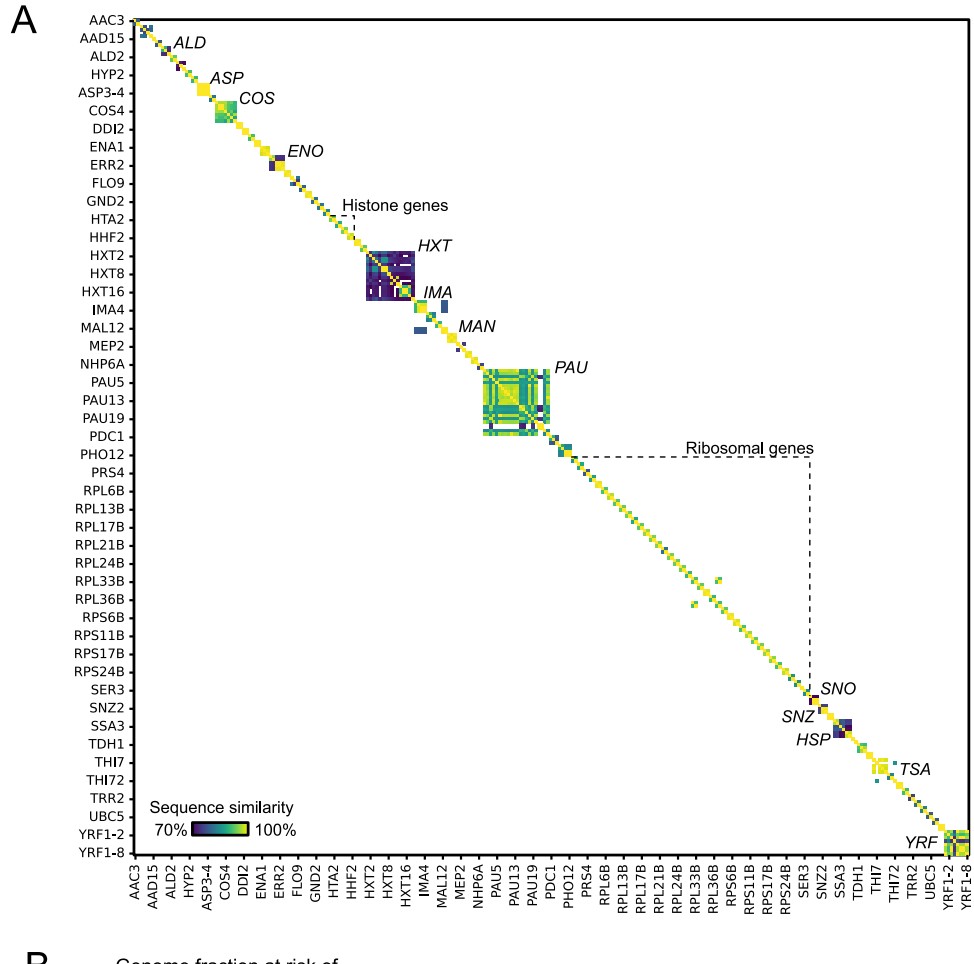

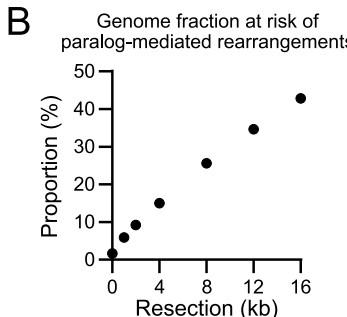

**Figure EV6. Analysis of the DNA sequence similarity of paralogous genes and their potential impact on genome stability (related to Fig. 6).**

(A) Pairwise similarity matrix for paralogous genes exhibiting >70% sequence similarity, obtained from Dataset EV2. (B) Fraction of the genome exposing >300 bp of at least one paralogous gene in (A) as a function of the resection length at randomly distributed DSBs.

