## [Peer Review File · The EMBO Journal]

Donor transcription suppresses D-loops in cis and promotes genome stability

Yasmina Djeghmoum and Aurèle Piazza

Corresponding author(s): Aurèle Piazza (aurele.piazza@ens-lyon.fr)

Review Timeline:

Submission Date:	21st Feb 25
Editorial Decision:	26th Mar 25
Revision Received:	27th Jun 25
Editorial Decision:	23rd Jul 25
Revision Received:	28th Jul 25
Accepted:	1st Aug 25

Editor: Hartmut Vodermaier

Transaction Report:

Dr. Aurèle Piazza
Ecole Normale Supérieure de Lyon
Laboratory of Biology and Modelling of the Cell
UMR5239
Lyon 69007
France

26th Mar 2025

Re: EMBOJ-2025-120563
Donor transcription regulates recombination in cis and promotes genome stability

Dear Dr. Piazza,

Thank you for submitting your structural study on transcription-recombination conflicts to The EMBO Journal. We sent it to three expert referees, who have now provided the reports copied below. As you will see, the referees express interest in your findings and acknowledge the sophisticated set-up of the study, but nevertheless retain certain reservations regarding the conclusiveness and insightfulness of some of the analyses.

Should you be able to strengthen the study in line with the major points of referees 2 and 3, we would be happy to consider a revised version further for publication. Please be reminded, however, that our single-major-revision-round policy makes it important to diligently respond to each referee point at the time of resubmission; therefore, please do not hesitate to contact me ahead of resubmission with any questions you may have in this regard. We would also be open to extension of the regular three-months revision period if needed; our 'scooping protection' (meaning that competing work appearing elsewhere in the meantime will not affect our considerations of your study) would of course remain valid also throughout such an extension.

Further information on preparing, formatting and uploading a revised manuscript can be found below and in our Guide to Authors. Thank you again for the opportunity to consider this work for The EMBO Journal, and I look forward to receiving your revised manuscript in due time.

Yours sincerely,

Hartmut Vodermaier

4) Each main and each Expanded View (EV) figure should be uploaded as individual production-quality files (preferably in .eps,

.tif, .jpg formats). For suggestions on figure preparation/layout, please refer to our Figure Preparation Guidelines: <http://bit.ly/EMBOPressFigurePreparationGuideline>

9) To facilitate reproducibility and cross-laboratory adoption of methodologies, please structure the Materials & Methods section as outlined in our guide to authors, including a completed Reagents and Tools Table that can be downloaded from our author guidelines as well (<https://www.embopress.org/page/journal/14602075/authorguide#structuredmethods>).

10) Digital image enhancement is acceptable practice, as long as it accurately represents the original data and conforms to community standards. If a figure has been subjected to significant electronic manipulation, this must be clearly noted in the figure legend and/or the 'Materials and Methods' section. The editors reserve the right to request original versions of figures and the original images that were used to assemble the figure. Finally, we generally encourage uploading of numerical as well as gel/blot image source data; for details see: embopress.org/page/journal/14602075/authorguide#sourcedata

At EMBO Press, we ask authors to provide source data for the main manuscript figures. Our source data coordinator will contact you to discuss which figure panels we would need source data for and will also provide you with helpful tips on how to upload and organize the files.

In the interest of ensuring the conceptual advance provided by the work, we recommend submitting a revision within 3 months (24th Jun 2025). Please discuss the revision progress ahead of this time with the editor if you require more time to complete the revisions. Use the link below to submit your revision:

Link Not Available

Referee #1:

In this study, the authors investigated the effect of donor transcription on the formation of strand invasion intermediates (D-loops) during homology-directed double-strand break repair. The experimental system consists of a recipient locus that is efficiently cleaved by HO endonuclease and a donor cassette located on a different chromosome (inter) or integrated on the same chromosome as the recipient locus (intra). Promoters of different strength were inserted upstream of the donor locus and D-loop formation was quantified by a proximity ligation-based assay. The authors found that transcription in the co-directional orientation inhibited nascent D-loop formation by ~10-fold using the inter-chromosomal donor, and by ~2-fold for the intra-chromosomal donor. In the head on transcriptional context, the repressive effect of transcription was still observed, but not to the same extent as the co-directional construct. They also demonstrated that the negative effect of donor transcription operates in cis, and is independent of R-loops or peripheral nuclear delocalization. Interestingly, the effect of transcription is more robust and independent of the known trans antagonists of D-loop formation, Mph1, Sgs1 and Srs2. Finally, the authors found that donor transcription reduced the frequency of complex multi-invasion recombination products.

Overall, the results are clearly presented and convincing. The observation that transcription has less of an impact on intra- compared with inter-chromosomal recombination suggests that it would be less of an impediment to sister chromatid recombination, the favored context in mitotically-dividing cells, but could be a mechanism to suppress mutagenic recombination between dispersed repeats. One caveat is that the assay used to measure inter-chromosomal recombination requires a tripartite

reaction and might be more affected than a simpler system involving two repeats.

Referee #2:

In the manuscript "Donor transcription regulates recombination in cis and promotes genome stability" Djeghmoum and Piazza investigate how homologous recombination (HR) is influenced by transcription of the donor locus. Specifically, they investigate how D-loop structures, which arise when single-stranded DNA at a double-strand break location invades the duplex DNA of a homologous donor sequence, are influenced by transcription of that donor sequence. To this end, they utilize a D-loop capture assay developed by earlier work of Piazza. Using this assay, they perform a battery of experiments that show it is specifically ongoing transcription of the donor locus, which disrupts the D-loops and not the RNA transcript, and RNA-DNA hybrid or simple stimulation of any of the known D-loop counteracting factors. Moreover, they can convincingly show that this mechanism may serve to counteract HR events by multi-invasion events, which otherwise can be a source of translocations. This is a strong paper describing an interesting and very well-designed set of experiments with clear results. It has a very clear logic and is easy to read. As such it is suited to for publication as soon as the authors address the following points.

Major Points:

#1 - One cannot understand from reading only this paper, how the D-loop capture assay works. I am not sure that all readers will make the effort to carefully check the earlier papers. As such, I would recommend that the authors include (e.g. as Supplemental Figure) an overview showing how the assay is done.

#2 - Fig. 2A-B - Here the authors manipulate transcription in the course of the experiment. Even though pCUP1 and pGAL1 are well studied promoters, it would be interesting to see the effects on transcription elongation (for example by Rpb1 ChIP) and correlate them to the effects on the D-loops.

#3 - Fig. 2C - The extent of D-loop suppression induced by transcription from pTDH3 seems to vary dependent on the specific set-up. Here, they only see a two-fold suppression compared to about 10-fold in Fig. 1. I suggest the authors make a short statement about these differences for example in the discussion.

#4 - Fig. 3B - The effect of the *tfiis* has both a low effect strength and a borderline significance. I think the authors need to tone down their statement: "On the contrary, the *tfiis* Δ mutant partly relieved the transcription-dependent D-loop inhibition (Fig. 3B)."

#5 - The paper convincingly shows that donor transcription leads to a disruption of D-loops. However, showing a direct influence on HR would further strengthen the paper.

Minor Points:

#1 - Fig. 4 - The labelling/legend for Co.-Dir./head-on collisions (filled and empty circles) is rather unclear. Why not label the graph directly.

#2 - There are some spelling/grammar mistakes: p. 6 "In order to addressed..."; p. 11 "we precise..."

#3 - Fig. 1D - "1kb plot" - graphs are connected to the origin of the diagram; "2kb plot" - graphs are not connected to the origin of the diagram;

#4 - Axis labels can be harmonized throughout the manuscript, as well as the use of "absolute D-loops" vs. "D-loops", for which figure legends show some inconsistencies.

#5 - Logic of citation is not clear to me on page 5 in the sentence: "We confirmed our improved DLC protocol (Reiz et al., 2022) [...] of transcription (Piazza et al., 2021) (Fig. S2C-D)."

#6 - I propose adding statistics to Fig. S3B. From the data spreading it is fair to assume this is an insignificant change, however it would support the claim that the *msh2* has no influence on D-loop suppression.

Referee #3:

In this manuscript, the authors used several D-loop reporters and the DLC assay to analyze the effects of donor DNA transcription on D-loop formation. They found that co-direction transcription of donor DNA efficiently inhibits D-loop formation. The inhibitory effect of donor transcription occurs in cis, is independent of RNA:DNA hybrids, and is less pronounced when transcription occurs in the head-on direction. This effect also appears to be independent of the known D-loop suppressing factors. Finally, donor transcription reduces multi-invasion events in chromosomes, suggesting that this mechanism may protect against chromosome rearrangement.

The experiments of this study are well designed and carefully executed, and the conclusions are largely convincing. One technical concern is on the effects of RNA on the DLC assay. Although the conclusions of this study are interesting, but the model seems to be somewhat incomplete. It is still not very clear how transcription inhibits D-loop formation mechanistically, and why the effects of co-directional and head-on transcription are different. If transcription directly disrupts D-loops or prevents the initiation events of D-loop formation, it would be helpful to show these effects experimentally.

1. Are the effects of transcription in donor DNA on D-loop formation specific to the DLC assay? If RNA transcript is crosslinked to the template DNA strand, does RNA affect the digestion and intramolecular ligation steps of DLC? This is an important technical issue to be addressed.

2. In fig. 3, it would be helpful to confirm the expected effects of RNH1 and mft1/tfiis mutants on the RNA:DNA hybrids in donor DNA.
3. The data showing that head-on transcription inhibits D-loop formation less efficiently than co-directional transcription is consistent with the possibility that RNA upstream of D-loops may interfere with the digestion or ligation step of DLC.
4. In the MIR assay, was the YS2 product detected? The process generating the YS2 product should be more similar to the D-loop reporters used for DLC. If the YS2 product was detected, was it affected by transcription of the S2 donor?

Dr. Aurèle Piazza
Ecole Normale Supérieure de Lyon
Laboratory of Biology and Modelling of the Cell
UMR5239
Lyon 69007
France

26th Mar 2025

Re: EMBOJ-2025-120563
Donor transcription regulates recombination in cis and promotes genome stability

Dear Dr. Piazza,

Thank you for submitting your structural study on transcription-recombination conflicts to The EMBO Journal. We sent it to three expert referees, who have now provided the reports copied below. As you will see, the referees express interest in your findings and acknowledge the sophisticated set-up of the study, but nevertheless retain certain reservations regarding the conclusiveness and insightfulness of some of the analyses.

Should you be able to strengthen the study in line with the major points of referees 2 and 3, we would be happy to consider a revised version further for publication. Please be reminded, however, that our single-major-revision-round policy makes it important to diligently respond to each referee point at the time of resubmission; therefore, please do not hesitate to contact me ahead of resubmission with any questions you may have in this regard. We would also be open to extension of the regular three-months revision period if needed; our 'scooping protection' (meaning that competing work appearing elsewhere in the meantime will not affect our considerations of your study) would of course remain valid also throughout such an extension.

Further information on preparing, formatting and uploading a revised manuscript can be found below and in our Guide to Authors. Thank you again for the opportunity to consider this work for The EMBO Journal, and I look forward to receiving your revised manuscript in due time.

Yours sincerely,

Hartmut Vodermaier

*** PLEASE NOTE: All revised manuscript are subject to initial checks for completeness and adherence to our formatting guidelines. Revisions may be returned to the authors and delayed in

their editorial re-evaluation if they fail to comply to the following requirements (see also our Guide to Authors for further information):

4) Each main and each Expanded View (EV) figure should be uploaded as individual production-quality files (preferably in .eps, .tif, .jpg formats). For suggestions on figure preparation/layout, please refer to our Figure Preparation Guidelines:

6) Please complete our Author Checklist, and make sure that information entered into the checklist is also reflected in the manuscript; the checklist will be available to readers as part of the Review Process File. A download link is found at the top of our Guide to Authors:

embopress.org/page/journal/14602075/authorguide

9) To facilitate reproducibility and cross-laboratory adoption of methodologies, please structure

the Materials & Methods section as outlined in our guide to authors, including a completed Reagents and Tools Table that can be downloaded from our author guidelines as well (<https://www.embopress.org/page/journal/14602075/authorguide#structuredmethods>).

10) Digital image enhancement is acceptable practice, as long as it accurately represents the original data and conforms to community standards. If a figure has been subjected to significant electronic manipulation, this must be clearly noted in the figure legend and/or the 'Materials and Methods' section. The editors reserve the right to request original versions of figures and the original images that were used to assemble the figure. Finally, we generally encourage uploading of numerical as well as gel/blot image source data; for details see: [embopress.org/page/journal/14602075/authorguide#sourcedata](https://www.embopress.org/page/journal/14602075/authorguide#sourcedata)

At EMBO Press, we ask authors to provide source data for the main manuscript figures. Our source data coordinator will contact you to discuss which figure panels we would need source data for and will also provide you with helpful tips on how to upload and organize the files.

Further information is available in our Guide For Authors:

In the interest of ensuring the conceptual advance provided by the work, we recommend submitting a revision within 3 months (24th Jun 2025). Please discuss the revision progress ahead of this time with the editor if you require more time to complete the revisions. Use the link below to submit your revision:

<https://emboj.msubmit.net/cgi-bin/main.plex?el=A5Ii6BFaw7A1CdBG3I3A9ftdINNMQZ6LT3EexJDJdsgY>

Referee #1:

In this study, the authors investigated the effect of donor transcription on the formation of strand invasion intermediates (D-loops) during homology-directed double-strand break repair. The experimental system consists of a recipient locus that is efficiently cleaved by HO endonuclease and a donor cassette located on a different chromosome (inter) or integrated on the same chromosome as the recipient locus (intra). Promoters of different strength were inserted upstream of the donor locus and D-loop formation was quantified by a proximity ligation-based assay. The authors found that transcription in the co-directional orientation inhibited nascent D-loop formation by ~10-fold using the inter-chromosomal donor, and by ~2-fold for the intra-chromosomal donor. In the head on transcriptional context, the repressive effect of transcription was still observed, but not to the same extent as the co-directional construct. They also demonstrated that the negative effect of donor transcription operates in cis, and is independent of R-loops or peripheral nuclear delocalization. Interestingly, the effect of transcription is more

robust and independent of the known trans antagonists of D-loop formation, Mph1, Sgs1 and Srs2. Finally, the authors found that donor transcription reduced the frequency of complex multi-invasion recombination products.

Overall, the results are clearly presented and convincing. The observation that transcription has less of an impact on intra- compared with inter-chromosomal recombination suggests that it would be less of an impediment to sister chromatid recombination, the favored context in mitotically-dividing cells, but could be a mechanism to suppress mutagenic recombination between dispersed repeats. One caveat is that the assay used to measure inter-chromosomal recombination requires a tripartite reaction and might be more affected than a simpler system involving two repeats.

We thank this reviewer for his/her positive appreciation of our work. We agree with this reviewer that a determination of the impact of transcription using a more conventional HR reporter would be a worthwhile addition to this study. Consequently, and in line with reviewer #2 and 3 comments, we evaluated the role of transcription on HR repair efficiency and canonical repair outcomes (NCO and CO) with two additional recombination assays. The first “ectopic” assay is derived from our experimental system. The second “allelic” assay is an adaptation of the *ade2* hetero-alleles system developed by the Symington lab (Ho et al. Mol. Cell 2010). The results are presented in **Fig. 5** and a new **Fig. 6**, respectively. Mainly, transcription suppressed ectopic repair ~6-fold without significantly distorting the NCO/CO ratio (**Fig. 5D-F**). The extent of this suppression is commensurate with that measured at the D-loop joint molecules and D-loop extension levels (~7-fold with 1 kb-long donor, **Fig. 1C, E**). Consequently, inhibition of product formation can be mainly, if not fully, ascribed to D-loop suppression. No such inhibition is observed in the allelic repair system (**Fig. 6A-C**). It suggests that broader homologies (a hallmark of allelic sites) can overcome the inhibition posed by transcription of a subset of the homology exposed by resection. Consequently, transcription is not an anti-recombination mechanism, but specifically an anti-ectopic recombination mechanism. We elaborate on these findings in an expanded discussion section (P13-14, **Fig. 6D-F**).

Referee #2:

In the manuscript "Donor transcription regulates recombination in cis and promotes genome stability" Djeghmoum and Piazza investigate how homologous recombination (HR) is influenced by transcription of the donor locus. Specifically, they investigate how D-loop structures, which arise when single-stranded DNA at a double-strand break location invades the duplex DNA of a homologous donor sequence, are influenced by transcription of that donor sequence. To this end, they utilize a D-loop capture assay developed by earlier work of Piazza. Using this assay, they perform a battery of experiments that show it is specifically ongoing transcription of the donor locus, which disrupts the D-loops and not the RNA transcript, and RNA-DNA hybrid or simple stimulation of any of the known D-loop counteracting factors. Moreover, they can convincingly show that this mechanism may serve to counteract HR events by multi-invasion events, which otherwise can be a source of translocations.

This is a strong paper describing an interesting and very well-designed set of experiments with clear results. It has a very clear logic and is easy to read. As such it is suited to for publication as soon as the authors address the following points.

We thank this reviewer for her/his positive appreciation of our work.

Major Points:

#1 - One cannot understand from reading only this paper, how the D-loop capture assay works. I am not sure that all readers will make the effort to carefully check the earlier papers. As such, I would recommend that the authors include (e.g. as Supplemental Figure) an overview showing how the assay is done.

We added a scheme describing the D-loop-Capture and D-loop-Extension assays in an updated **Fig. S1A**.

#2 - Fig. 2A-B - Here the authors manipulate transcription in the course of the experiment. Even though pCUP1 and pGAL1 are well studied promoters, it would be interesting to see the effects on transcription elongation (for example by Rpb1 ChIP) and correlate them to the effects on the D-loops.

The transcriptional level at the donor under the control of all promoters has been checked by Rpb1 ChIP-qPCR and/or by measuring the level of the resulting RNA, and were presented in the original **Fig. S1A-B** (for *pDMC1*, *pLYS2*, *pTDH3*, and *pGAL1*) and in **Fig. S2A** (for pCUP1 10 minutes after addition of various concentration of copper). We showed an anti-correlation between D-loop levels with RNA levels in the original **Fig S1D** (new **Fig. S1E**). We now also show the anti-correlation between D-loop levels and the Rpb1 ChIP-qPCR levels in an updated **Fig. S1E**, also significant (Pearson $R^2=0.9899$, $p=0.0005$).

The donor RNA levels increase by at least an order of magnitude 10 minutes after copper addition when the donor is placed under the control of the *pCUP1* promoter (new **Fig. S2A**). These RNA levels are similar to that measured for the *pTDH3* and *pGAL1* promoters (new **Fig. S1C**). Following this reviewer advice, we added the anti-correlation between RNA levels and D-loop levels measured at varying copper concentration in a new **Fig. S2B** (Pearson's $R^2=0.9267$, $p=0.0373$), which further supports that D-loop suppression depends on the transcriptional level at the donor site.

The kinetics of expression shutoff from *pGAL1* upon glucose addition to galactose-containing media has previously been determined to occur within minutes (for instance see RNA levels in Fig 5 in Johnston .. Pexton MCB 1994; 10.1128/mcb.14.6.3834-3841.1994) and was not reproduced here.

#3 - Fig. 2C - The extent of D-loop suppression induced by transcription from pTDH3 seems to vary dependent on the specific set-up. Here, they only see a two-fold suppression compared to about 10-fold in Fig. 1. I suggest the authors make a short statement about these differences for example in the discussion.

This difference partly relates to homology length. Indeed, in **Fig. 2C** the donor is 2 kb-long, while the 10-fold suppression in **Fig. 1C** is obtained with a 1 kb-long donor. Increasing length of homology appears to partly counteract the effect of transcription, with 2 kb of homology leading to a 2- to 3-fold D-loop suppression (comparison in **Fig. 1E**). The proximity of the break and the

donor site in the “intra” context may also partly contribute to overcome the effect of inhibitory effect of transcription. We noted this point in the discussion P11 (now P13):

“Consistently, we noted that long homologies (Fig. 1E) and DSB-donor spatial proximity (Fig. 2C), two hallmarks of the sister chromatid, partly counteracted the effect of transcription.”

#4 - Fig. 3B - The effect of the *tfiisΔ* has both a low effect strength and a borderline significance. I think the authors need to tone down their statement: "On the contrary, the *tfiisΔ* mutant partly relieved the transcription-dependent D-loop inhibition (Fig. 3B)."

Indeed, we acknowledge this borderline effect of *tfiisΔ*, and provided the exact p-value (0.055) for critical assessment by the readers. We rephrased the results on P6 as follows

*“The *tfiis* mutant actually exhibited a modestly reduced transcription-dependent D-loop inhibition, at the limit of statistical significance ($p=0.055$, two-tailed unpaired Mann-Whitney test, **Fig. 3B**). Given the role of *TFIIS* in promoting transcription elongation by *RNA PolIII* (Sigurdsson et al, 2002; Zatreanu et al, 2019), this observation suggests that D-loop suppression requires efficient transcription across the donor.”*

#5 - The paper convincingly shows that donor transcription leads to a disruption of D-loops. However, showing a direct influence on HR would further strengthen the paper.

We acknowledge that our study is primarily focused on the strong effect of transcription at the earliest DNA joint molecule intermediate of HR. Yet we also provide evidence that this early regulation exerts downstream effects on repair steps and products:

- In **Fig. 1F** we showed that transcription caused a commensurate ~7-fold decrease in the amount of D-loop extension products.
- In **Fig. 4F-H** (now **Fig. 5B-C**) we measured the frequency of a HR-dependent repair event (*i.e.* the MIR translocation).

Consequently, the data presented go beyond the sole effect of transcription in regulating D-loop intermediates.

We nonetheless agree with this reviewer that a determination of the impact of transcription on other, more conventional HR outcomes would be a worthwhile addition to this study. Consequently, and in line with reviewer #1 and 3 comments, we evaluated the role of transcription on HR repair efficiency and canonical repair outcomes (NCO and CO) with two additional recombination assays. The first “ectopic” assay is derived from our experimental system. The second “allelic” assay is an adaptation of the *ade2* hetero-alleles system developed by the Symington lab (Ho et al. Mol. Cell 2010). The results are presented in **Fig. 5** and a new **Fig. 6**, respectively. Mainly, transcription suppressed ectopic repair ~6-fold without significantly distorting the NCO/CO ratio (**Fig. 5D-F**). The extent of this suppression is commensurate with that measured for D-loop joint molecules and D-loop extension (~7-fold with 1 kb-long donor, **Fig. 1C, E**). Consequently, inhibition of product formation can be mainly, if not fully, ascribed to D-loop suppression. No such inhibition is observed in the allelic repair system (**Fig. 6A-C**). It suggests that broader homologies (a hallmark of allelic sites) can overcome the inhibition posed by transcription of a subset of the homology exposed by resection. Consequently, transcription is

not an anti-recombination mechanism, but specifically an anti-ectopic recombination mechanism. We elaborate on these findings in an expanded discussion section (P13-14, **Fig. 6D-F**).

Minor Points:

#1 - Fig. 4 - The labelling/legend for Co.-Dir./head-on collisions (filled and empty circles) is rather unclear. Why not label the graph directly.

We chose this representation because it allows to keep the graph labels consistent between the D-loop values in panels B/D and the ratios in panels C/E.

#2 - There are some spelling/grammar mistakes: p. 6 "In order to addressed..."; p. 11 "we precise..."

Thanks for noticing, we fixed these errors.

#3 - Fig. 1D - "1kb plot" - graphs are connected to the origin of the diagram; "2kb plot" - graphs are not connected to the origin of the diagram;

This is because we only determined the D-loop levels prior to DSB induction (0 hours) with the 2 kb construct (n=1). In all the cases tested so far D-loops are undetectable prior to DSB induction (Piazza et al. Mol Cell 2019; Reitz et al. Genes Dev. 2023). We removed this data point and re-centered the graph between 2 and 8 hours, as in panel E.

#4 - Axis labels can be harmonized throughout the manuscript, as well as the use of "absolute D-loops" vs. "D-loops", for which figure legends show some inconsistencies.

Thanks, we harmonized text and figures.

#5 - Logic of citation is not clear to me on page 5 in the sentence: "We confirmed our improved DLC protocol (Reiz et al., 2022) [...] of transcription (Piazza et al., 2021) (Fig. S2C-D)."

The first citation refers to the improved DLC protocol, described in (Reitz 2022) while the second citation refers to the "~20-fold preference at the intra over the inter donor" determined in (Piazza 2021) with the original DLC protocol (Piazza 2019).

#6 - I propose adding statistics to Fig. S3B. From the data spreading it is fair to assume this is an insignificant change, however it would support the claim that the *msh2Δ* has no influence on D-loop suppression.

Indeed, we performed the statistical test between the WT and *msh2Δ* strains, which turned out to be non-significant in both the *pDMC1* and *pTDH3* contexts. We now indicate this lack of significance in the legend of **Fig. S3B**. Whenever pertinent, the lack of statistical significance is now indicated in the figure legends.

Referee #3:

In this manuscript, the authors used several D-loop reporters and the DLC assay to analyze the effects of donor DNA transcription on D-loop formation. They found that co-direction transcription of donor DNA efficiently inhibits D-loop formation. The inhibitory effect of donor transcription occurs in cis, is independent of RNA:DNA hybrids, and is less pronounced when transcription occurs in the head-on direction. This effect also appears to be independent of the known D-loop suppressing factors. Finally, donor transcription reduces multi-invasion events in chromosomes, suggesting that this mechanism may protect against chromosome rearrangement.

The experiments of this study are well designed and carefully executed, and the conclusions are largely convincing. One technical concern is on the effects of RNA on the DLC assay. Although the conclusions of this study are interesting, but the model seems to be somewhat incomplete. It is still not very clear how transcription inhibits D-loop formation mechanistically, and why the effects of co-directional and head-on transcription are different. If transcription directly disrupts D-loops or prevents the initiation events of D-loop formation, it would be helpful to show these effects experimentally.

We thank this reviewer for his/her positive assessment of our work. Indeed, we identified a transcription-dependent HR control at the D-loop level and provided a first delineation of its mechanistic origin, yet we still lack a detailed understanding as the inhibition/disruption process. In the discussion section “Putative mechanism of transcription-mediated D-loop suppression” on P11, we speculate that active passage of the RNA PolIII at the donor provides the energy required to branch-migrate D-loops until they are fully unwound, and that the initiation or efficiency of this migration depends on the D-loop junction encountered by RNA PolIII:

“RNA PolIII is a processive directional molecular motor threading along dsDNA, a process facilitated by TFIIS (Gnatt et al, 2001; Kettenberger et al, 2004; Charlet-Berguerand et al, 2006). Such active translocation may dissociate already formed D-loops by mechanically migrating its strand exchange junctions, an energetically neutral reaction. The fact that transcription mainly causes loss of co-directional D-loops (i) suggests that it does not solely act by preventing the upstream step of Rad51-ssDNA NPF binding to dsDNA at that site, and (ii) makes it unlikely that D-loop dissociation is primarily mediated by topological changes at the donor. It instead suggests that the prioritization of transcription over the synaptic steps of HR depends on the type of DNA strand exchange junctions encountered by RNA PolIII, and/or of the presence of HR proteins decorating them relative to an incoming RNA PolIII. Consistently, RNA PolIII obtained from HeLa cells extract could traverse a model co-linear D-loop substrate in vitro, while a head-on D-loop was a roadblock (Pipathsouk et al, 2017). The precise structural basis for this orientation-dependent behavior, at the heart of the prioritization between the two processes, remains to be determined.”

We feel, however, that the precise dissection of the mechanism of D-loop suppression by RNA PolIII requires in-depth biochemical and structural studies; an undertaking out of the scope of the present work.

1. Are the effects of transcription in donor DNA on D-loop formation specific to the DLC assay? If RNA transcript is crosslinked to the template DNA strand, does RNA affect the digestion and intramolecular ligation steps of DLC? This is an important technical issue to be addressed.

We evaluated the effect of donor transcription on recombination with other methods than the DLC assay. First, we could show on **Fig. 1F** that transcription causes a ~10-fold decrease in D-loop extension; determined with an assay that does not involve crosslinking (rationale see new **Fig. S1A**). Second, using a genetic assay we showed that donor transcription inhibits MIR. Third, we could show that, in a competitive donor system, transcription of one donor could lead to an increase of D-loops at the non-transcribed donor (**Fig. 2C-D**). Finally, transcription inhibits ectopic recombination ~6-fold, as scored by cell survival and molecular detection of NCO and CO repair products (new **Fig. 5D-F**, no crosslink). These four independent evidences refute this reviewer’s technical concern.

We also want to clarify that the restriction sites used in DLC and DLE are not located within the promoters or the transcribed donor. The EcoRI site near the donor used for DLC is located 85 bp away from the promoter, in a region that remains constant in all constructs used in this study (see genetic construct scheme in **Fig S1A**).

To further assuage this reviewer's concern, we nonetheless performed additional controls on the DLC experiments, which are now part of **Fig. S1D**. First, we controlled for the digestion of the EcoRI restriction site located upstream of the donor used in the DLC assay with the two main promoters used in this study, *pDMC1* and *pTDH3*. Transcription of the donor did not cause a significant change in digestion efficiency. Second, to address the effect of transcription on the intramolecular ligation efficiency, we determined the circularization efficiency of a DNA fragment as a function of its transcriptional status. The transcribed donor itself could not be scored in this manner, as it lacks a compatible restriction site downstream of the donor. We used as a proxy the circularization of a 2.7 kb fragment containing the *GAL3* gene, in galactose- and glucose-containing media. Transcription caused a minor decrease in the intra-molecular ligation efficiency, from 26.1% (without) to 22.8% (with). This minor difference cannot explain the >7-fold effect of transcription on D-loop levels.

2. In fig. 3, it would be helpful to confirm the expected effects of *RNH1* and *mft1/tfiis* mutants on the RNA:DNA hybrids in donor DNA.

The various conditions used to manipulate RNA:DNA hybrids abundance in yeast have been selected based on extensive literature (*mft1/THO* complex: (Huertas & Aguilera, 2003; Chávez *et al*, 2000; Gómez-González *et al*, 2011; Bonnet *et al*, 2017; Appanah *et al*, 2020; Penzo *et al*, 2023; Mangione *et al*, 2025) *tfiis*: (Zatreanu *et al*, 2019; Kay *et al*, 2024; Duardo *et al*, 2024) and *RNH1* over-expression: (Huertas & Aguilera, 2003; Wahba *et al*, 2011; Nguyen *et al*, 2017; Appanah *et al*, 2020; Sanders *et al*, 2025). These different conditions consistently failed to affect D-loop levels in ways suggestive of an involvement of RNA:DNA hybrids.

We now provide the RT-qPCR controls for *RNH1* over-expression in a new **Fig. S3A**. This system achieves a ~1,500-fold over-expression above endogenous *RNH1* transcript levels.

3. The data showing that head-on transcription inhibits D-loop formation less efficiently than co-directional transcription is consistent with the possibility that RNA upstream of D-loops may interfere with the digestion or ligation step of DLC.

Please see response to point 1 for additional controls performed on the digestion and ligation steps of DLC protocol.

Please note that RNA is not a substrate for any of the enzymes involved in the DLC reaction.

4. In the MIR assay, was the YS2 product detected? The process generating the YS2 product should be more similar to the D-loop reporters used for DLC. If the YS2 product was detected, was it affected by transcription of the S2 donor?

We performed an experiment similar to that proposed by this reviewer in **Fig. 1F**, in which we quantified D-loop extension (rationale in new **Fig. S1A**; no crosslink involved). We found that transcription caused a ~7-fold reduction in extended D-loops at highly-transcribed donors.

Transcription-mediated D-loop disruption thus causes a commensurate decrease in the downstream extension product.

We understand that this reviewer wishes to see effects of donor transcription on more conventional HR repair outcome, in line with other comments by reviewer 1 and 2. Accordingly, we evaluated the role of transcription on HR repair efficiency and canonical repair outcomes (NCO and CO) with two additional recombination assays. The first “ectopic” assay is derived from our experimental system. The second “allelic” assay is an adaptation of the *ade2* hetero-alleles system developed by the Symington lab (Ho et al. Mol. Cell 2010). The results are presented in **Fig. 5** and a new **Fig. 6**, respectively. Mainly, transcription suppressed ectopic repair ~6-fold without significantly distorting the NCO/CO ratio (**Fig. 5D-F**). The extent of this suppression is commensurate with that measured at the D-loop joint molecules and D-loop extension levels (~7-fold with 1 kb-long donor, **Fig. 1C, E**). Consequently, inhibition of product formation can be mainly, if not fully, ascribed to D-loop suppression. No such inhibition is observed in the allelic repair system (**Fig. 6A-C**). It suggests that broader homologies (a hallmark of allelic sites) can overcome the inhibition posed by transcription of a subset of the homology exposed by resection. Consequently, transcription is not an anti-recombination mechanism, but specifically an anti-ectopic recombination mechanism. We elaborate on these findings in an expanded discussion section (P13-14, **Fig. 6D-F**).

Dr. Aurèle Piazza
Ecole Normale Supérieure de Lyon
Laboratory of Biology and Modelling of the Cell
UMR5239
Lyon 69007
France

23rd Jul 2025

Re: EMBOJ-2025-120563R
Donor transcription suppresses D-loops in cis and promotes genome stability

Dear Dr. Piazza,

Thank you for submitting your revised manuscript to The EMBO Journal. It has now been re-reviewed by original referees 1 and 2, who were both fully satisfied with the revisions. We shall therefore be happy to accept the study for publication, as soon as a few remaining editorial issues have been addressed:

- Please adjust the order of the manuscript sections: Title page with complete author information, Abstract, Keywords, Introduction, Results, Discussion, Methods, Data Availability, Acknowledgements, Disclosure and Competing Interests Statement, References, Main Figure Legends, Tables, Expanded Figure Legends.
- On the abstract page of the manuscript, please include 4-5 general keyword terms to enhance searchability.
- Please rename the Conflict of Interest section into "Disclosure and Competing Interests Statement", in accordance with our updated Guide to Authors (<https://www.embopress.org/competing-interests>)
- As we are switching from a free-text author contribution statement towards a more formal statement based on Contributor Role Taxonomy (CRediT) terms, please remove the present Author Contribution section and instead specify each author's contribution(s) directly in the Author Information page of our submission system during upload of the final manuscript. See <https://casrai.org/credit/> for more information.
- Please carefully go through the reference list and make sure that each reference is complete with citation year, volume, and page/locator numbers (currently missing for several of them).
- Please make sure to upload our author checklist fully completed (noting also the general information sections).
- Please remove the Reagents and Tools table from the main article file, it is sufficient that it is uploaded as a separate text file.
- Please remove mention to "supplementary" information from the manuscript text. The legends for Figures EV1-6 should be prefaced "Expanded View Figure Legends". EV Tables should have their legends within the file, either on top (as in Tables EV2&3) or in a separate "Legends" tab of the spreadsheet file. Moreover, Table EV4 should be renamed to Dataset EV2, and also have its title/legend in a separate Legends tab. Finally, Dataset EV1 also requires legends/information, in this case ideally in the form of a Readme text file included in the ZIP archive.
- Please provide suggestions for a short 'blurb' text prefacing and summing up the conceptual aspect of the study in two sentences (max. 250 characters), followed by 3-5 one-sentence 'bullet points' with brief factual statements of key results of the paper; they will form the basis of an editor-written 'Synopsis' accompanying the online version of the article. Please also upload a synopsis image, which can be used as a "visual title" for the synopsis section of your paper. The image should be in PNG or JPG format, and please make sure that it remains in the modest dimensions of (exactly) 550 pixels wide and 300-600 pixels high.
- Finally, during routine pre-acceptance checks, our data editors have raised the following queries regarding figures, data, and legends, which I would ask you to address (ideally using the Track Changes option):
 1. Please note that the exact p values are not provided in the legends of figures 1C, 2C, D; 3C, D; 4B, D; 5B, E, F.
 2. Please indicate the statistical test used for data analysis in the legends of figures 1C, 2C, D; 3B-D; 4B, D; 5B, E, F.
 3. Please note that information related to n is missing in the legends of figures 1C, D, E, F; 2C, D; 3A, B, C, D, E; 4B-E; 5B, C, E, F; 6B, C; EV1 A-F; EV2 A, B, E, G; EV3 B-C; EV4 A, EV5A
 4. Please note that the error bars are not defined in the legends of figures EV2 B, E; EV3 A-C; EV4 A, EV5A

I am returning the manuscript to you for a final round of minor revision, solely to allow you to make these modifications and upload the revised files. Once we will have received them, we should be ready to swiftly proceed with formal acceptance and production of the manuscript.

With kind regards,

Hartmut

9) To facilitate reproducibility and cross-laboratory adoption of methodologies, please structure the Materials & Methods section as outlined in our guide to authors, including a completed Reagents and Tools Table that can be downloaded from our author guidelines as well (<https://www.embopress.org/page/journal/14602075/authorguide#structuredmethods>).

10) Digital image enhancement is acceptable practice, as long as it accurately represents the original data and conforms to community standards. If a figure has been subjected to significant electronic manipulation, this must be clearly noted in the figure legend and/or the 'Materials and Methods' section. The editors reserve the right to request original versions of figures and the original images that were used to assemble the figure. Finally, we generally encourage uploading of numerical as well as gel/blot image source data; for details see: embopress.org/page/journal/14602075/authorguide#sourcedata

Further information is available in our Guide For Authors:

In the interest of ensuring the conceptual advance provided by the work, we recommend submitting a revision within 3 months (21st Oct 2025). Please discuss the revision progress ahead of this time with the editor if you require more time to complete the revisions. Use the link below to submit your revision:

Link Not Available

Referee #1:

In this study, the authors show convincingly that transcription of a donor locus inhibits Rad51-mediated D-loop formation. This inhibitory effect acutely reduces inter-chromosomal recombination between short repeats but is not observed for allelic sites that share extensive regions of flanking homology. Their findings have important implications for maintenance of genome stability by suppression of recombination between highly transcribed paralogous genes.

The authors have responded well to the critiques from the initial review with additional experiments to address the role of transcription in suppressing recombination in ectopic and allelic contexts. I have no further comments/concerns.

Referee #2:

The revised version of "Donor transcription suppresses D-loops in cis and promotes genome stability" addresses my previous comments in full. I congratulate the authors on a very interesting study, which in my opinion is ready to be published in EMBO J.